# A thermophilic chemolithoautotrophic bacterial consortium suggests a mutual relationship between bacteria in extreme oligotrophic environments

Yuri Pinheiro[1,8], Fabio Faria da Mota [2,8], Raquel S. Peixoto[3,4], Jan Dirk van Elsas[5], Ulysses Lins [1], Jorge L. Mazza Rodrigues [6] & Alexandre Soares Rosado [3,4,7 ✉]

A thermophilic, chemolithoautotrophic, and aerobic microbial consortium (termed carbonitroflex) growing in a nutrient-poor medium and an atmosphere containing $N_2$, $O_2$, $CO_2$, and CO is investigated as a model to expand our understanding of extreme biological systems. Here we show that the consortium is dominated by *Carbonactinospora thermoautotrophica* (strain StC), followed by *Sphaerobacter thermophilus*, *Chelatococcus* spp., and *Geobacillus* spp. Metagenomic analysis of the consortium reveals a mutual relationship among bacteria, with *C. thermoautotrophica* StC exhibiting carboxydotrophy and carbon-dioxide storage capacity. *C. thermoautotrophica* StC, *Chelatococcus* spp., and *S. thermophilus* harbor genes encoding CO dehydrogenase and formate oxidase. No pure cultures were obtained under the original growth conditions, indicating that a tightly regulated interactive metabolism might be required for group survival and growth in this extreme oligotrophic system. The breadwinner hypothesis is proposed to explain the metabolic flux model and highlight the vital role of *C. thermoautotrophica* StC (the sole keystone species and primary carbon producer) in the survival of all consortium members. Our data may contribute to the investigation of complex interactions in extreme environments, exemplifying the interconnections and dependency within microbial communities.

[1] Institute of Microbiology, Federal University of Rio de Janeiro, Rio de Janeiro, Brazil. [2] Computational and Systems Biology Laboratory, Oswaldo Cruz Institute, FIOCRUZ, Rio de Janeiro, Brazil. [3] Red Sea Research Center (RSRC), King Abdullah University of Science and Technology (KAUST), Thuwal 23955-6900, Saudi Arabia. [4] Computational Bioscience Research Center (CBRC), King Abdullah University of Science and Technology (KAUST), Thuwal 23955-6900, Saudi Arabia. [5] Microbial Ecology, Groningen University, Groningen, the Netherlands. [6] Department of Land, Air, and Water Resources, University of California Davis, Davis, CA, USA. [7] Bioscience Program, Biological and Environmental Sciences and Engineering Division (BESE), King Abdullah University of Science and Technology (KAUST), Thuwal, Saudi Arabia. [8] These authors contributed equally: Yuri Pinheiro, Fabio Faria da Mota. ✉ email: alexandre.rosado@kaust.edu.sa

All biogeochemical cycles involve microbial biosynthesis, and microbes can substantially promote carbon dioxide sequestration and nitrogen fixation[1–3], which are essential societal goals[4]. However, the mechanisms associated with biomass formation, which involves their complex microbiomes, intricate interactions, and functions, have not been completely elucidated; and understanding these mechanisms remains a major challenge[5–7].

Oligotrophic ecosystem-based microbial consortia may serve as models for the investigation of complex biotechnological processes and thus expand our understanding of biological systems. Microorganisms cooperate with each other to cope with various conditions, such as limited nutrient availability and/or stress[8]; however, extreme environmental conditions may lead to highly interdependent interactions in microbial consortia. The key biogeochemical processes underlying life in these systems may also be compartmentalized. A better understanding of this interdependence and the interaction modes driving microbial community interactions[9] may lead to the discovery of novel pathways or genes[10]. This understanding may eventually lead to novel biotechnological applications[11,12] through the optimization and engineering of microbial consortia to produce biomolecules, biofuels, and carbon sequestration[3,13,14], and through the use of the principles of microbial ecology[15]. These advances may contribute to mitigation of crucial issues, such as climate change and population growth[11,16,17].

Extremophiles such as carboxydotrophs can evolve rapidly to adjust to the dynamic changes that may occur under extreme conditions, thereby making them good sources of novel bioproducts[18]. Studies of oligotrophic consortia developing under extreme conditions are ideal to elucidate the organic chemistry of (early) life on Earth or other planets/moons in the solar system[19,20].

In this study, a thermophilic and oligotrophic bacterial consortium was successfully enriched from soil; we investigated the microbial composition of the bacterial growth, to determine whether the extreme selective conditions shaped the enriched bacterial consortium, and elucidate the collaborations among members of the consortium to adjust to harsh conditions. The consortium—termed carbonitroflex (CNF)—was enriched from a soil sample with a decadal history of repeated grass burning (long-term exposure to high temperatures and $CO_2$ from the fires).

We used a polyphasic approach to characterize the microbial consortium. Metagenomic analysis of the consortium facilitated the construction of a model that explained interactions and mutual relationships among bacteria to ensure their survival under restrictive environmental conditions. The proposed metabolic model could identify the primary producer (breadwinner) and other groups (collaborators and cheaters) in this network.

## Results

### Morphological characterization.
The CNF thermophilic bacterial consortium was obtained at 55 °C from the soil collected from underneath a pile of burned grass, which had a different chemical composition from that of a soil sample collected from a nearby forest. The soil from the burn site had markedly higher P and K contents, pH, and a C:N ratio (Supplementary Fig. 1C). The consortium was able to grow and form floating white pellicles in a mineral medium without a source of organic carbon and nitrogen, and no growth was observed for any of the negative controls applied. The initial floating white pellicles were observed 15 days after the beginning of isolation attempts, in only one of the inoculated soil samples (Fig. 1a). This first culture was then incubated for an additional 15 days. The resulting growth appeared as less-concentrated pellicles of floating cell agglomerates on both media, solid N-FIX supplemented with clinoptilolite (Fig. 1b) and N-FIX liquid medium (Fig. 1c). This culture was then transferred every 14 days to triplicate vials (each time) for >3 years. The cell agglomerates (floating white pellicles) were consistently observed after 3–21 days in all replicates for >3 years, with electron microscopy. Filamentous bacteria containing intracellular spores were observed in association with a smaller number of single-celled cocci and bacilli. These microbes appeared to be closely attached to each other (Fig. 1d), with bacilliform bacteria being attached beneath the filamentous cells. Several spore-like structures directly bound to the bacterial cell filaments were also observed (Fig. 1e). Images obtained using SEM revealed the presence of a cell lining, which was possibly a bridge of unknown material linking the cells (Fig. 1f). Remarkably, highly symmetrical bacterial microcompartments (BMCs)/carboxysome-like structures in the cellular cytoplasm were also noted (Fig. 2). Interestingly, regularly shaped clusters were noted in the cellular cytoplasm. These diamond-shaped clusters appeared in several images and in different sizes (probably due to the block cutting at different heights). Genomic data indicated (see below) that these might be carboxysome-like structures, which are bacterial microcompartments that concentrate carbon-fixing enzymes and play an essential role in the carbon fixation process[21].

### Composition and stability of the CNF consortium.
The consistent stability of the consortium was evaluated using DGGE community profiling and 16 S ribosomal RNA gene barcoding, over a period of approximately 3 years of cultivation (44 transfers) at variable intervals of months. During this period, the DNA of the consortium was extracted at different times (after 5, 12, 16, 36, and 44 transfers), and DGGE and Illumina sequencing were performed to compare the profile of the bacterial community, based on the ribosomal gene *rrs*. The results suggested that despite small fluctuations and changes in band intensities (even after normalization) and bacterial composition, the bacterial core of the consortium remained stable (Supplementary Fig. 2A, B).

After 3 years, >99.5% of the 1,032,456 metagenomic reads obtained from CNF were taxonomically assigned. The results indicated that the consortium comprised 4 bacterial species (Fig. 3). There was no evidence of the presence of other prokaryotes or fungi. The most abundant reads (approximately 80%) observed in the consortium metagenome belonged to an Actinobacterium that was identified as closely related to *Carbonactinospora thermoautotrophica*[22], followed by a Chloroflexum related to *Sphaerobacter* spp. (9% of reads), a Proteobacterium (8% of reads) related to *Chelatococcus* spp., and a Firmicute (2% of reads) related to *Geobacillus* spp. (Fig. 3a). The phylogeny of the Barrnap-extracted gene *rrs* is shown in Fig. 3b. Notably, two different contigs aligned with *C. thermoautotrophica* sequences, indicating that this strain (*C. thermoautotrophica* St_consortium [StC]) harbored two taxonomically independent *rrs* genes (StCopy1 and StCopy2). A high degree of difference (>6%) was observed between the *rrs* genes of *C. thermoautotrophica* StC.

Table 1 shows the CheckM output, indicating the degree of completeness of the MAGs used in complementation analyses. All MAGs demonstrated > 95% completeness. A cutoff of 1% was used, thereby indicating that any read belonging to any other species was <1% of the total reads. This finding was supported by plating, where no growth was observed. Using BRIG and DNAPlotter software and BLASTN program, the contigs belonging to each consortium member were represented in a circular form and mapped to the reference WGS (UBT1 and H1

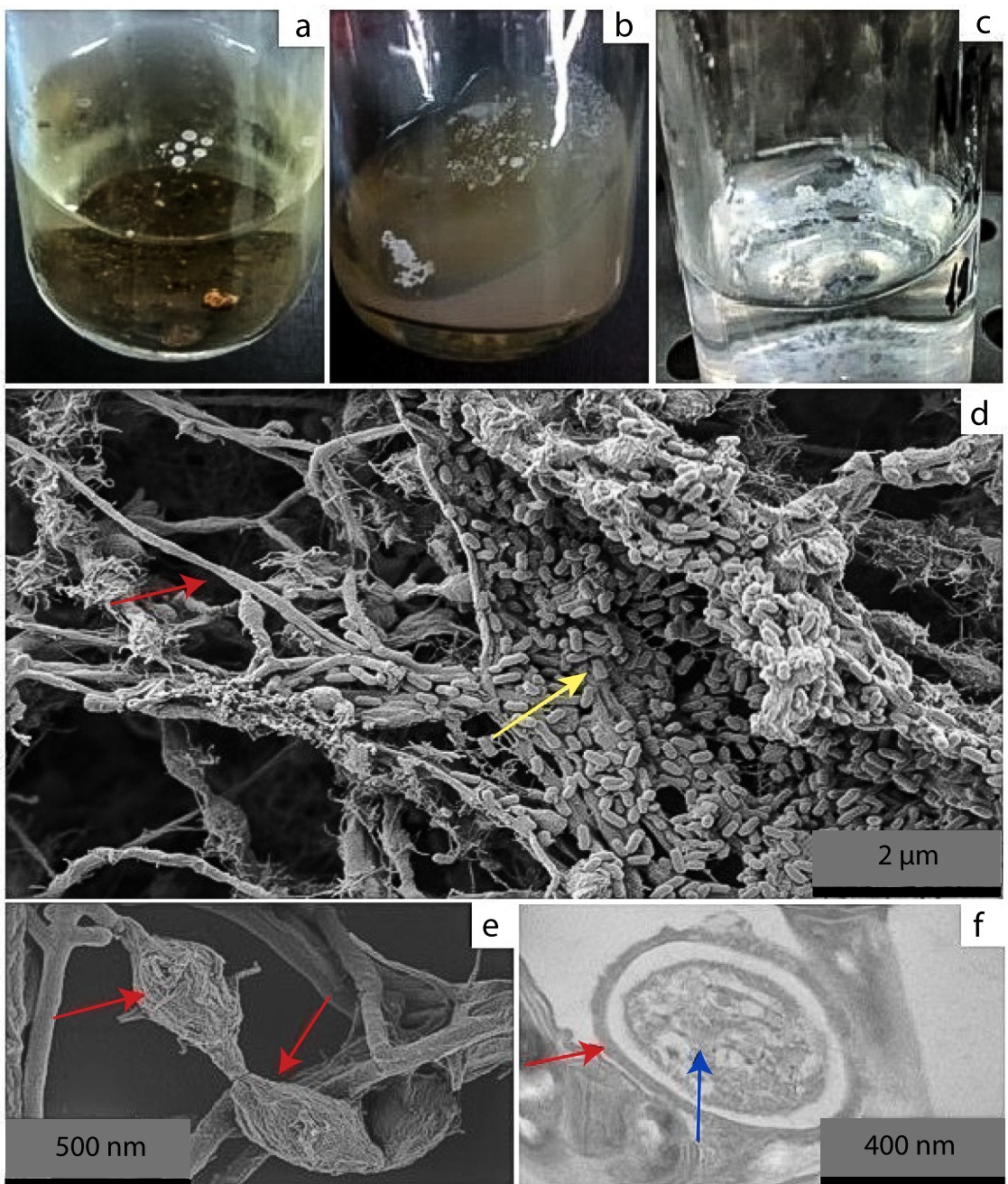

**Fig. 1 Morphological overview of the chemolithoautotrophic consortium (carbonitroflex).** Initial growth from soil samples. **a** Colonies growing in mineral medium supplemented with $NH_4Cl$; **b** growth on N-FIX medium supplemented with clinoptilolite; **c** growth on N-FIX medium filled with synthetic air, CO, and $CO_2$; **d** confocal electron photomicrograph of carbonitroflex grown in N-FIX medium without nitrogen-fixation sources (magnification 11,210×) showing non-filamentous bacteria (yellow arrow) deposited on actinobacterial filamentous growth (red arrow); **e** scanning electron photomicrograph depicting the details of the spore-like structure (red arrows) (magnification 51,000×); and **f** transmission electron photomicrograph (magnification 71,000×) showing the vacuoles (blue arrows), well-delimited membrane, and cell wall (red arrow).

for *C. thermoautotrophica*, DSM207045 for *Sphaerobacter* spp., DSM18167 for *Chelatococcus* spp., and DSM13240 for *Geobacillus* spp.) (Fig. 4a–c). Metagenomic reads were also mapped (Fig. 3d) for the previously isolated and sequenced *Geobacillus* sp. LEMMY01 genome[23] (Fig. 4d).

Different solid culture media and the loop-depletion technique were used to isolate the different members of the CNF consortium. Despite being a minority (2% of reads) in the consortium, *Geobacillus* spp. was the sole member of the consortium that could be isolated because it could be easily

cultured in R2A medium. However, no growth of this pure culture was obtained in N-FIX. Its genome was previously sequenced by ref. [23].

**CNF consortium metabolism.** Enzymes responsible for the aerobic respiration of molecular $H_2$ and CO were only encoded in three members of the consortium: *C. thermoautotrophica*, *Chelatococcus* spp., and *Sphaerobacter* spp. *C. thermoautotrophica* possessed hydrogenases belonging to groups 1 and 2a, whereas

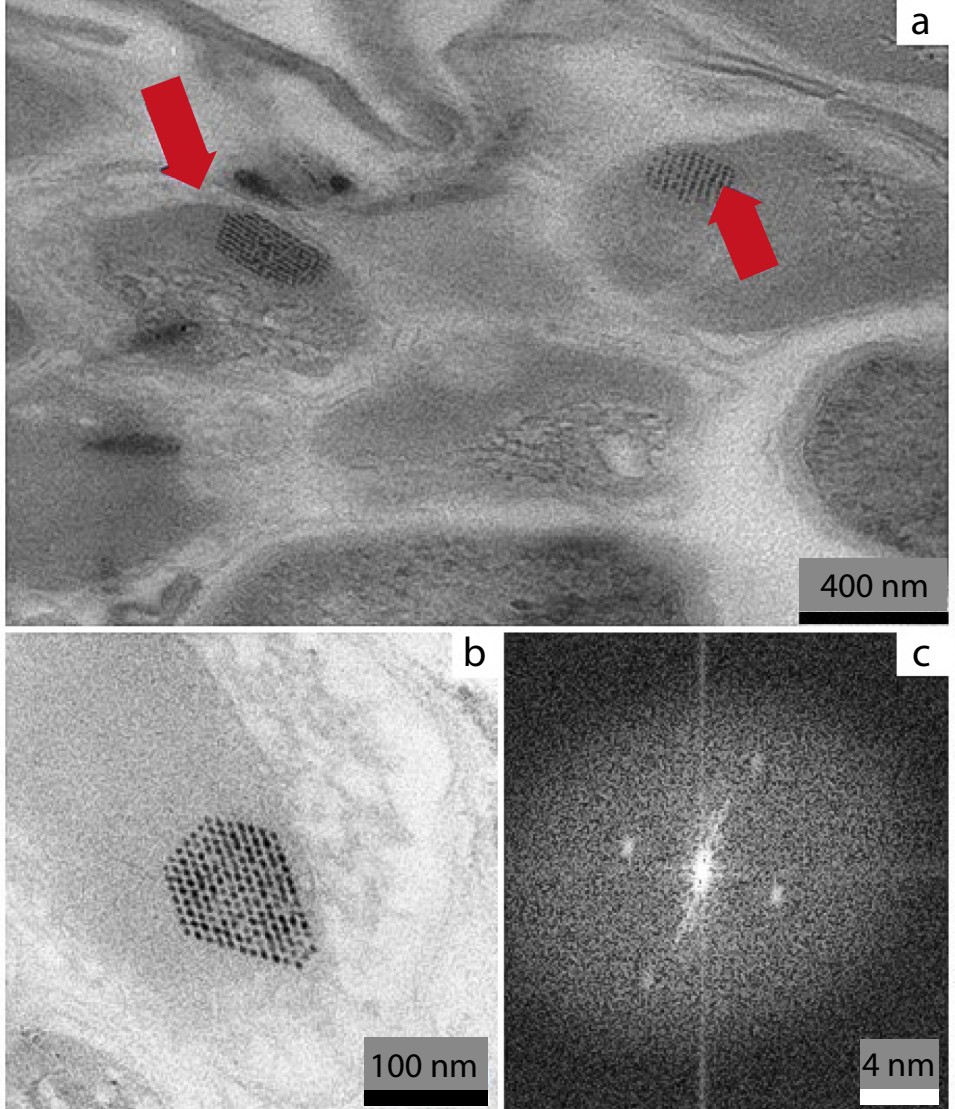

**Fig. 2 Transmission electron photomicrographs (magnification 71,000×) showing the highly symmetrical bacterial microcompartments (BMCs)/ carboxysome-like structures. a** Red arrows indicate carboxysome-like structures in the cellular cytoplasm of *Carbonactinospora thermoautotrophica* StC; **b** a close view of a carboxysome- like structure; and **c** The Fast Fourier Transform (FFT) image of the carboxysome-like structure, which is a mathematical representation of the spatial frequencies of the image[25], showing that the StC carboxysome-like structure has a periodic organization with a well-ordered and highly symmetrical pattern.

*Sphaerobacter* spp. possessed those belonging to group 1. More specifically, *C. thermoautotrophica* and *Sphaerobacter* spp. MAGs harbored genes encoding group 1 h [NiFe]-hydrogenases (*hhyL* and *hhyS*) (Fig. 5, Supplementary Data 2) as well as type I [MoCu]-carbon monoxide dehydrogenases (*coxL, coxS,* and *coxM*); moreover, despite the absence of hydrogeneses, *Chelatococcus* spp. also harbored genes encoding enzymes required for CO respiration. In addition, no $CO_2$ fixation genes were detected in *Chelatococcus* spp. and *Sphaerobacter* spp.

*C. thermoautotrophica* harbors genes that enable obtaining energy from CO and/or $H_2$ oxidation. This bacterium can grow aerobically, with $H_2$ or CO as its sole energy and electron source[24]; it uses the energy derived from CO and/or $H_2$ oxidation to support both aerobic respiration and carbon fixation through the CBB cycle. Being the sole member containing genes supporting $CO_2$ fixation, *C. thermoautotrophica* StC appeared to be responsible for carbon fixation, as it harbored the type IE ribulose-1,5-bisphosphate carboxylase (RuBisCO) enzyme, a

member of the Calvin–Benson–Bassham (CBB) cycle. Its shell and accessory enzymes were also found in the StC genome, such as CcmL, annotated as a carbon dioxide-concentration mechanism protein. Indeed, structures of bacterial microcompartments (BMC-like) related to carboxysomes were observed by transmission electron microscopy (Fig. 2a, b). The Fast Fourier Transform, which is a mathematical representation of the spatial frequencies of the image[25], showed that the StC carboxysomes have a periodic organization with a well-ordered pattern. The distance between each arrangement is between 7 and 8 nm (Fig. 2c).

Regarding the nitrogen cycle, the KEGG Pathway database and MEGAN5 software showed that none of the reads obtained from the metagenomes was related to previously characterized nitrogenase-encoding genes, such as *nifD* (nitrogenase molybdenum-iron protein alpha chain [EC:1.18.6.1]), *nifH* (nitrogenase iron protein), and *nifK* (nitrogenase molybdenum-iron protein beta chain [EC:1.18.6.1])[26]. Moreover, PCR

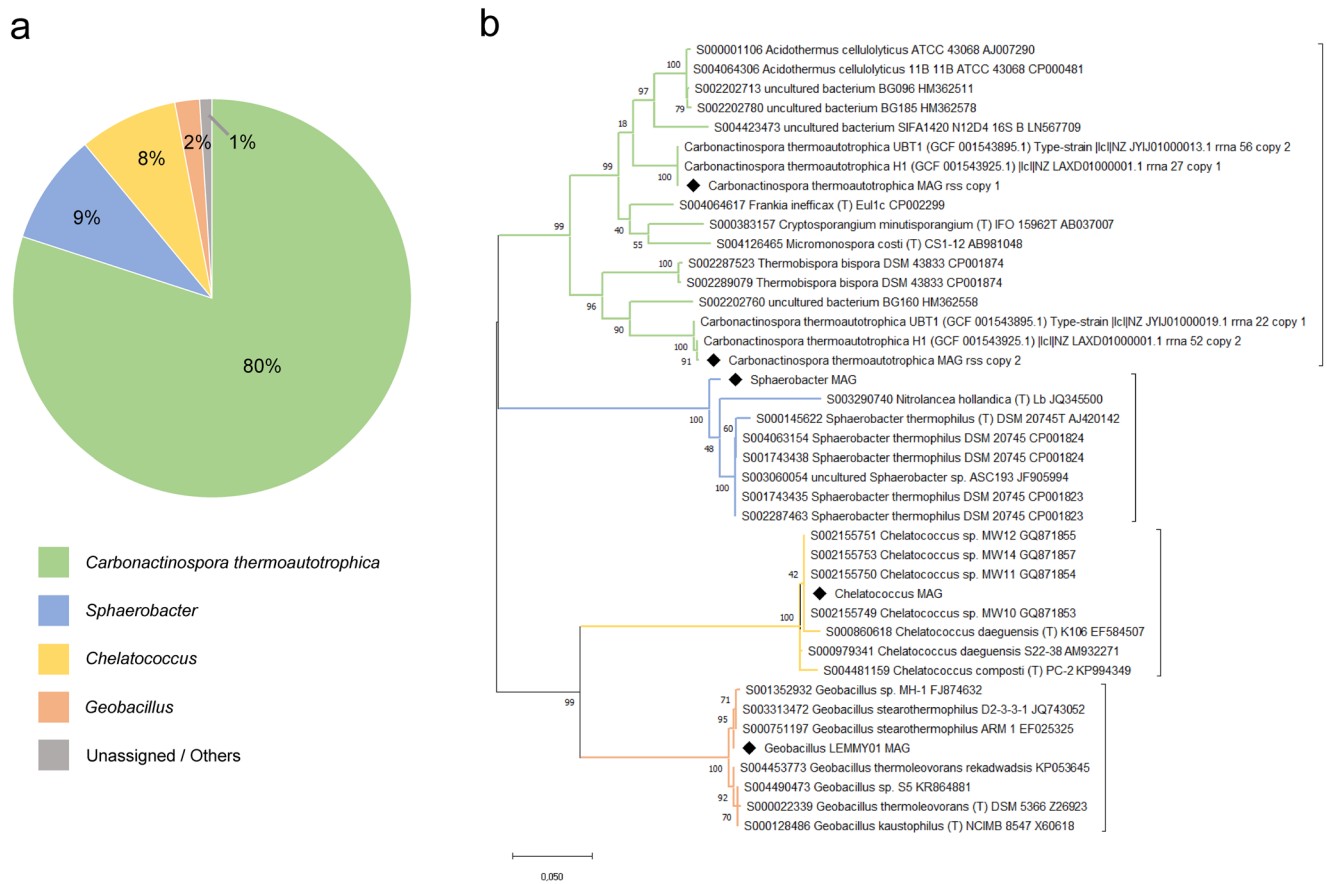

**Fig. 3 Carbonitroflex metagenome composition. a** Taxonomic composition and **b** molecular phylogenetic analysis of the carbonitroflex consortium based on the gene *rrs* (16 S); the maximum-likelihood method and Tamura–Nei distance model[57] were used to infer the evolutionary history. The tree with the maximum log probability is shown. The percentage of bootstrap trees in which the associated taxa clustered in this manner is shown adjacent to the branches. The *rrs* copies filtered from MAGs are highlighted with a diamond shape. The initial trees for the heuristic search were automatically obtained by applying the neighbor-joining and BioNJ algorithms to a matrix of estimated pairwise distances, using the maximum composite likelihood approach, and selecting the topology with the maximum logarithmic likelihood value. The tree is drawn to scale, with branch lengths proportional to the Tamura–Nei distance. The analysis included 32 nucleotide sequences. All positions containing gaps and missing data were eliminated. Evolutionary analyses were performed using MEGA7[56].

**Table 1 CheckM output table showing the estimated contamination and completeness from the recovered MAGs and the *Geobacillus* LEMMY01 isolate.**

| Genome | Marker Lineage | Reference Genomes | Markers | Marker Sets | Completeness (%) | Contamination (%) |
|---|---|---|---|---|---|---|
| Carbonoactinospora | o__Actinomycetales | 488 | 310 | 185 | 99.46 | 2.07 |
| Chelatococcus | o__Rhizobiales | 92 | 481 | 319 | 95.42 | 12.44 |
| Sphaerobacter | k__Bacteria | 924 | 159 | 107 | 96.26 | 0.1 |
| Geobacillus.LEMMY01 | f__Bacillaceae | 128 | 561 | 183 | 99.45 | 0.55 |

Genome statistics were determined according to the presence or absence of marker genes and the expected colocalization of these genes compared with the marker lineage[64].

amplification using degenerate *nifH* primers provided negative results in the metagenomic DNA, although positive results were obtained for *H. seropedicae* (positive control).

*C. thermoautotrophica* StC harbored the *nirBD* genes encoding nitrite reductase in bacteria; meanwhile, the *Sphaerobacter* spp. MAG harbored denitrifying genes related to N₂O reduction (*nos* gene encoding a multicopper N₂O reductase associated with the reduction of $N_2O$ to $N_2$) and denitrification (*nirK* gene encoding a periplasmic copper-containing nitrite reductase involved in the reduction of $NO_2^-$ to NO).

Nitrogen-labeled assays were performed to determine whether the growth of the CNF consortium was independent of exogenous combined nitrogen. The use of a $^{15}N_2$ isotope tracer showed no

enrichment of $^{15}N$ in the CNF consortium. Only the positive control (*H. seropedicae*) incorporated $^{15}N_2$ into the biomass under the test conditions. No growth of *E. coli* or no inoculation (negative controls) was observed in the nitrogen-free medium (Table 2).

The combined metabolic versatility of the carbon, nitrogen, and sulfur pathways and genes associated with the oxygen and hydrogen metabolism present in the CNF consortium sorted using MAG is shown in Fig. 5 (see also Supplementary Data 2). Overall, carbon fixation may be performed by *C. thermoauto-trophica* StC, which harbors genes encoding RuBisCO enzymes, in addition to *coxL*, *coxM*, and *coxL* encoding CO dehydrogenase and formate oxidase; in addition, it possesses *fdhA*, *fdhB*, and *fdhC* encoding formate dehydrogenases, which appear to be

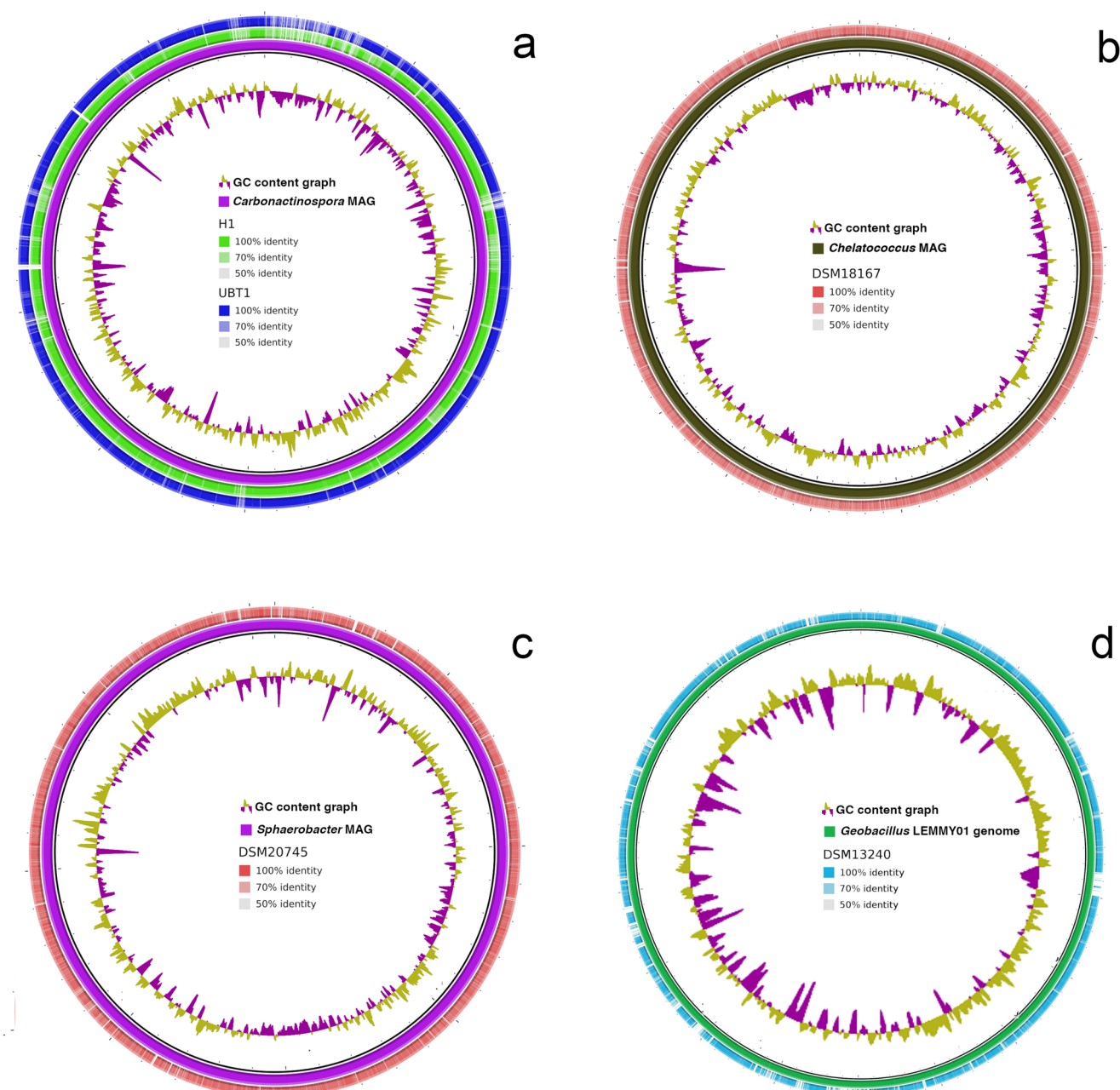

**Fig. 4 Circular representation of the comparison between carbonitroflex MAGs and their closest reference strains. a** Actinomycete isolated from the carbonitroflex consortium (*Carbonactinospora thermoautotrophica* St_consortium) and strains UBT1 and H1; **b** *Chelatococcus* spp. consortium strain and the reference strain DMS18167; **c** *Sphaerobacter thermophilus* from the consortium and the reference strain DSM20745; and **d** *Geobacillus* spp. strain LEMMY01 and the reference strain DSM13240. More-intense colors represent higher BLASTN identities; GC content is indicated in the inner circles.

involved in the catalytic functions of CO dehydrogenase and formate oxidase. Moreover, as noted above, *C. thermoautotrophica* StC possessed groups 1 and 2a [NiFe]-hydrogenases. The functionality in this case was inferred based on homology; therefore, these functional assignments must be confirmed in future studies.

Consistent with the PCR data, the presence of nitrogen-fixation pathways could not be determined. Remarkably, the nitrogen cycle-related function in the CNF consortium comprised denitrification in *Sphaerobacter* spp. MAG using $N_2O$ reductase (encoded by *nosZ* and *nosD*) and nitrite reductase (encoded by *nirK* and *nirD*) and in *C. thermoautotrophica* StC MAG using the enzymes encoded by *nirB* and *nirD* (responsible for the reduction

of nitrite to $NH_4$ via an assimilation pathway). These data suggest that $NO_2$ produces nitrogen, possibly in the form of $NH_4$ via nitrite reduction, which is then further incorporated into glutamate. The presence of proteins involved in nitrification was not predicted in the MAGs of any consortium member. Partial and/or complete open reading frames for all important orthologous proteins involved in denitrification were predicted in the MAGs of *Sphaerobacter* spp. and *Chelatococcus* spp. However, in both metagenomes, crucial protein-coding genes required for the final reduction of NO to $N_2$ were missing, thereby resulting in the incomplete mapping of denitrification.

A possible explanation for the different metabolic pathways driving the consortium interactions is presented in Fig. 6. We

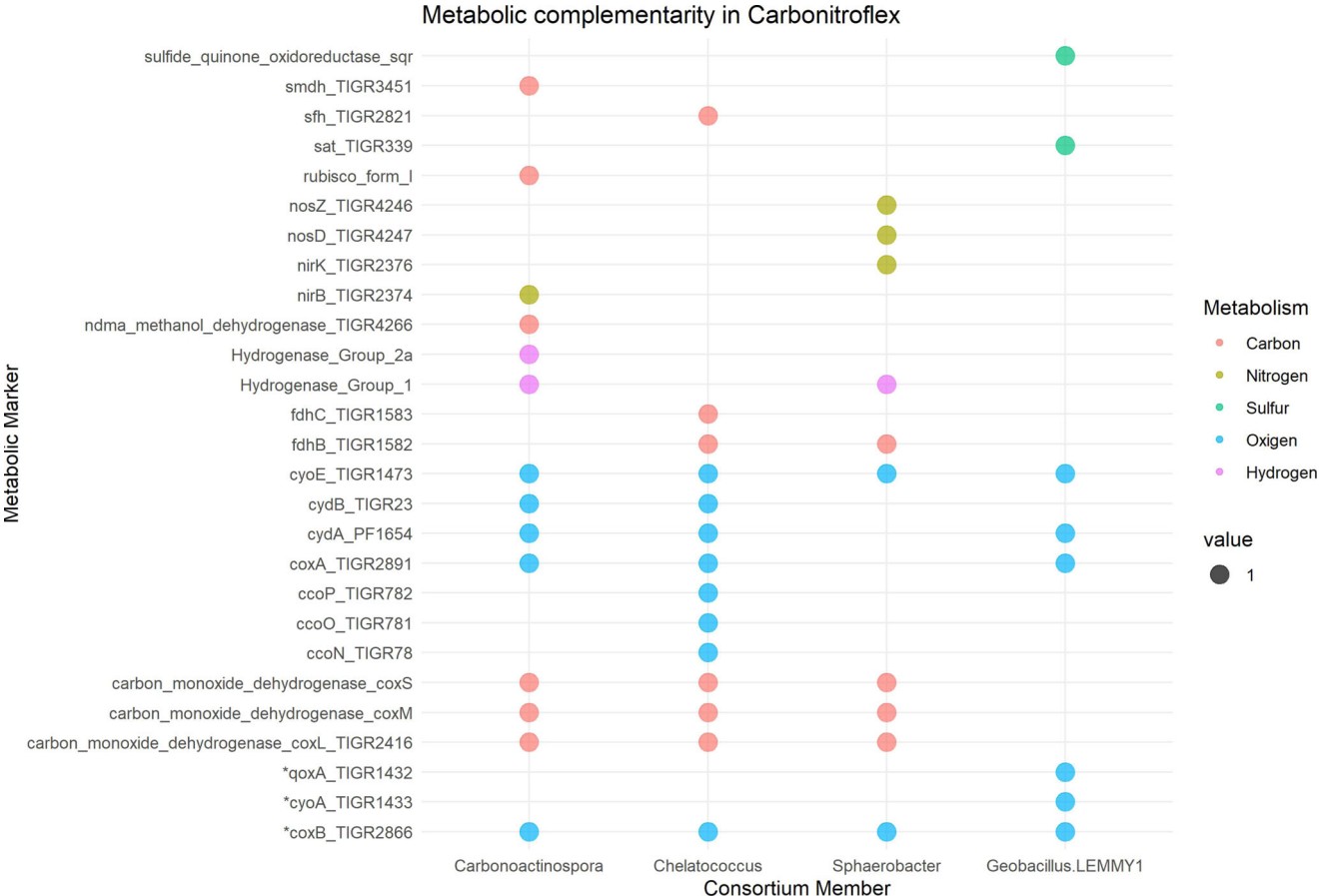

**Fig. 5 Overall metabolic functions across the MAGs and *Geobacillus* spp. genome comprising the carbonitroflex consortium.** Metabolic markers of carbon, nitrogen, sulfur, oxygen, and hydrogen cycles were identified via HMM search using the metabolisHMM tool. Metabolic markers with * represent genes with high homology. (See also Supplementary Data 2).

**Table 2 Growth of the Carbonitroflex consortium (CNF) or control strains in the presence of $^{15}N_2$ gas.**

| Sample/Strain | Nutrition | $^{15}N_2$ (% of Available $N_2$) | Atom % 15N | n* |
|---|---|---|---|---|
| *H. seropedicae* ATCC 35892 | Malic acid | 16 | 3.2 (±4.0) | 3 |
| *H. seropedicae* ATCC 35892 | Malic acid/$NH_4$ | 16 | 0.65 (±0.25) | 3 |
| CNF | CO | 16 | 0.36 (±0.01) | 3 |
| CNF | CO/$NH_4$ | 16 | 0.36 (±0.02) | 3 |
| *E. coli* BL21 | Glucose/$NH_4$ | 16 | 0.35 (±0.01) | 3 |
| *H. seropedicae* ATCC 35892 | Malic acid | 48 | 3.96 (±3.2) | 3 |
| *H. seropedicae* ATCC 35892 | Malic acid/$NH_4$ | 48 | 0.6 (±3.2) | 2 |
| CNF | CO | 48 | 0.36 (±0.03) | 3 |
| CNF | CO/$NH_4$ | 48 | 0.36 (±0.04) | 3 |
| *E. coli* BL21 | Glucose/$NH_4$ | 48 | 0.35 (±0.04) | 3 |

*n = number of replicates.

suggest that *C. thermoautotrophica* StC is the chief player involved in maintaining viable CNF under extreme thermophilic and oligotrophic conditions. Therefore, an arrangement termed the breadwinner hypothesis was proposed to explain the relationship among the members of this model consortium. The concept is discussed below.

## Discussion

The long history of repetitive grass burning of vegetal material may have facilitated selection of thermophilic carboxydotrophic bacteria that comprise the unique metabolisms observed in our CNF consortium. Several sources, including biomass burning, are responsible for the emission of CO, $CO_2$, $H_2$, NOx, $N_2O$, and $NH_3$[27], which may have enhanced the microorganisms that are able to grow without any source of organic carbon and nitrogen by consuming gaseous atmosphere (CO, $N_2$, and $O_2$). The persistence of the less-abundant taxa observed over the continuum replication process (>3 years) indicated that the organisms are either complementing each other's metabolism or are benefiting from the secondary metabolites from different players.

Microscopic analysis revealed that this consortium comprised gram-positive filamentous bacteria with spores, which were associated with a small number of cocci and bacilli. No other

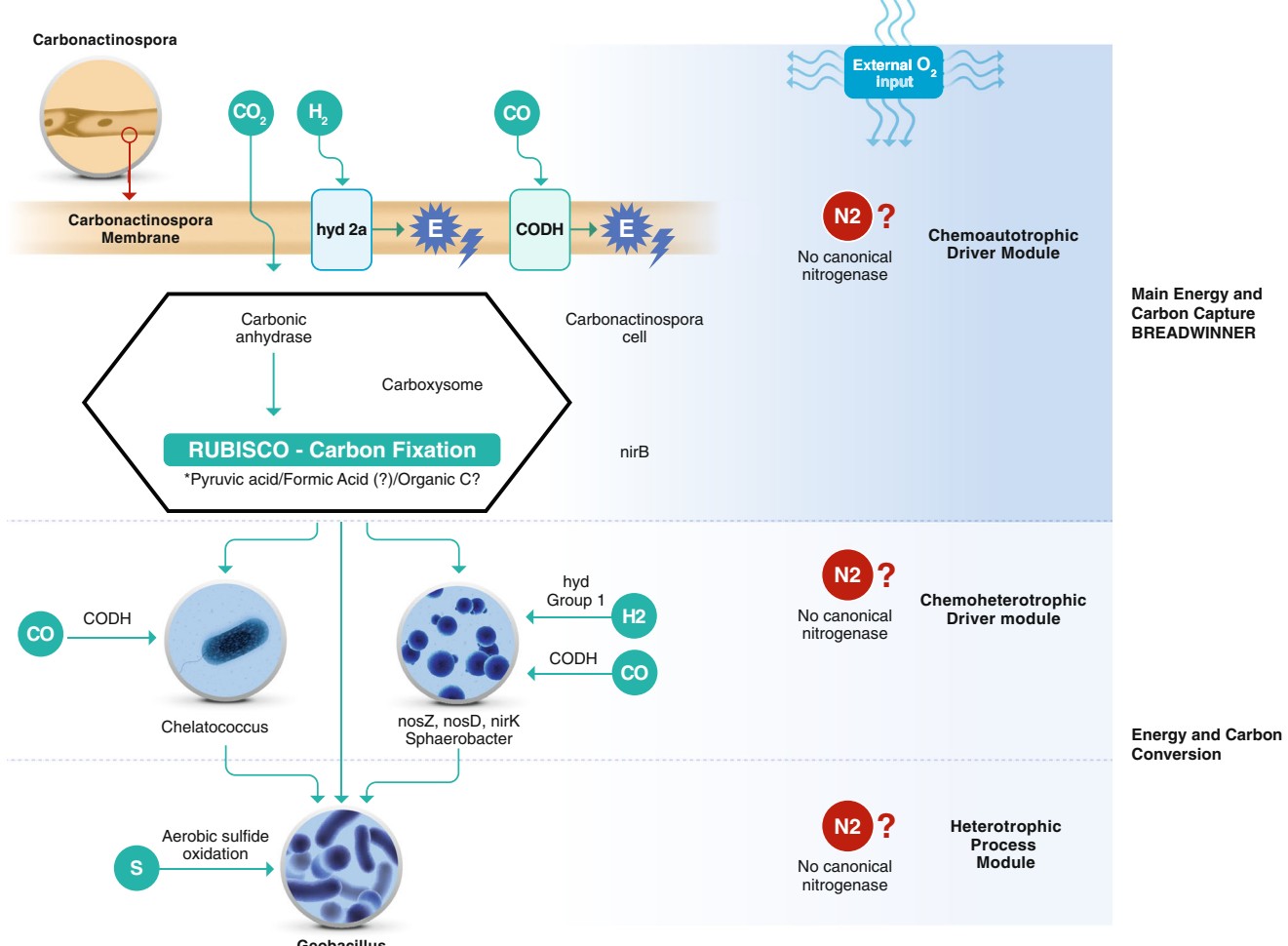

**Fig. 6 Proposed metabolic pathways that drive the carbonitroflex consortium interactions.** Main energy and carbon capture are provided by *Carbonactinospora thermoautotrophica* StC (the breadwinner and sole member capable of carbon fixation), whereas the fixed carbon integrated into the consortium is used by *Sphaerobacter thermophilus*, *Chelatococcus* spp., and *Geobacillus* spp.

microbes were detected, and no fungal mycelia were noted. Moreover, the results demonstrated that the filamentous StC bacterial strain in the consortium contains carboxysome-like structures. Carboxysomes are a family of BMCs present in cyanobacteria and some proteobacteria that encapsulate the primary $CO_2$-fixing enzyme, RuBisCO, within a virus-like polyhedral protein shell[21]. This indicates a degree of compartmentalization for carbon fixation in the CNF consortium.

In addition to the physiological and morphological evaluations, MAG and genomic data confirmed the presence of carboxysome genes in *C. thermoautotrophica* StC. Although an uncommon phenomenon, the large difference observed (>6%) between *rrs* copies in *C. thermoautotrophica* StC has also been reported in other Actinobacteria, and supports our taxonomic identification because this is a common characteristic of *C. thermoautotrophica*[22,24].

The other members present in the bacterial culture were taxonomically related to *Chelatococcus* spp., *Geobacillus* spp., and *Sphaerobacter* spp., which are bacterial genera that are known to contain thermophilic species[18].

Following phylogenetic analysis, the metabolic potential of the consortium members was investigated. All four organisms harbored genes encoding proteins that facilitated aerobic respiration; this finding was consistent with the results obtained in our experimentally created aerated growth conditions. Meanwhile, genes required for the oxidation of organic carbon (heterotrophic

metabolism) were only detected in *Sphaerobacter* spp., *Chelatococcus* spp., and *Geobacillus* spp. The high-affinity lineages identified in *C. thermoautotrophica* StC support its ability to scavenge $H_2$ and CO present in trace concentrations in the atmosphere. In addition, *C. thermoautotrophica* StC MAG harbored genetic information required for carboxydotrophy (*coxS*, *coxM*, and *coxL*) and autotrophy through type IE RuBisCO, which is a member of the CBB cycle. Although no genes related to $CO_2$ fixation were detected in *Chelatococcus* spp. and *Sphaerobacter* spp., they possessed complete pathways for oxidizing CO and could obtain energy from this trace gas. Notably, the expression of group 1 hydrogenase detected in *C. thermoautotrophica* StC and *Sphaerobacter* spp. and group 2a detected in *C. thermoautotrophica* StC could facilitate the use of $H_2$, another trace gas, as an energy source by these microbes. Cumulatively, our data indicated that *C. thermoautotrophica* StC primarily supplies carbon to the microbial consortium through carbon fixation. The energy for the system is provided by *C. thermoautotrophica* StC, *Chelatococcus* spp., and *Sphaerobacter* spp., as they appear to be able to use CO as an energy source. The metabolic compartmentalization of these species, along with the inability to isolate individual components among them, indicates some level of interdependence (at least from some members) and collaboration among the taxa, whereas *Geobacillus* spp. probably benefits from organic compounds produced by the other members in such oligotrophic systems.

Regarding nitrogen uptake, alkaline soils, such as the soil from which we obtained our samples, tend to have a higher availability of ammonia[28], which could be a source of nitrogen for CNF members in their natural environment. Nevertheless, for long-term survival under highly adverse conditions, such as our N-FIX medium, the consortium may depend on nitrogen fixation and/or a high nitrogen scavenging capacity of one or more members of the consortium that are able to utilize trace amounts of $NH_3$. However, in the present study, the consortium including four strains showed no classical nitrogenase-encoding genes. More-over, we detected no nitrogen-fixation activity using the isotope tracer ($^{15}N_2$) in the experimental setup. No evidence for (classi-cal) $N_2$ fixation was observed; therefore, the ability of this con-sortium to grow in the N-FIX medium suggests the existence of an efficient mechanism to obtain the nitrogen for biomass generation.

The possibility that the gaseous mixture was contaminated with trace amounts of $NH_4$ (despite the 99% purity guaranteed by the company) could not be completely eliminated. Moreover, trace contamination from air could be present in the medium because $NH_3$ dissolves extremely rapidly in water.

Yoshida et al.[29] reported that *Rhodococcus erythropolis* N9T-4, an extreme oligotrophic actinomycete isolate, can grow in mini-mal medium without a nitrogen source. The nitrogen oligotrophy of N9T-4 involves the strong expression of an ammonium transporter gene (*amtB*); the authors suggested that N9T-4 can utilize a trace amount of atmospheric ammonia as a nitrogen source. Therefore, nitrogen scavenging is another parsimonious explanation for the growth of microbes because it has previously been demonstrated in other microbes surviving in a nitrogen-free medium[24,29]. However, the presence of super-scavenger bacteria that can thrive under such conditions is unlikely to explain the growth of members of the entire consortium. Moreover, the use of clinoptilolite (a powerful chelator of trace ammonia) and Noble agar did not inhibit the growth of the consortium. Although nitrogen scavenging cannot be completely ruled out because of the hypothetical presence of unknown metabolic paths (for example, the recent description of a novel mechanism for biological nitrogen fixation by ref. [30]), we believe that the strict measures employed in the present study would possibly generate insufficient nitrogen (if any) to support bacterial growth and thus would only encourage the growth of true diazotrophs on the N-FIX medium.

The absence of a classical nitrogen fixation pathway raises questions regarding the strict nitrogen budget in this extremely harsh oligotrophic environment and the need for the residing extremophiles to grow under nitrogen scarcity. One of the pri-mary concerns in the current study was the growth of the con-sortium without a reduced nitrogen source ($NH_3$ or organic nitrogen). Theoretically, necromass or a viral shunt could sustain some amount of minimal growth[31]. In fact, Shoemaker et al.[32] demonstrated that for closed systems, necromass recycling con-tributed to the maintenance of energy-limited cells and also facilitated some reproduction. However, explaining the growth of the consortium based only on a viral shunt, cannibalism, or necromass use appears highly unreasonable. Therefore, we sug-gest that the consortium fixed $N_2$ via an unknown mechanism or that at least one extremely efficient scavenger responsible for the increased biomass for all members was present in the consortium, perhaps providing necromass or organic compounds (including nitrogen sources) to the others.

Furthermore, the presence of proteins involved in nitrification could not be predicted. Moreover, the metagenomes of *Sphaer-obacter* spp. and *Chelatococcus* spp. exhibited a lack of crucial genes encoding proteins involved in the final reduction of NO to $N_2$; this resulted in the incomplete mapping of denitrification.

However, incomplete denitrification pathways are common in extremophiles, particularly thermophiles[33]. Typically, deni-trification requires low oxygen levels and high nitrate levels, which differ from the growth conditions in the present study.

From the consortium, only *Geobacillus* spp. (LEMMY01) could be isolated, using a low-nutrient Reasoner's 2 A medium (R2A), which is among the media used most often for the isolation and growth of oligotrophic, environmental bacteria. However, the same strain was incapable of growing axenically in an N-FIX medium with a gaseous atmosphere. Despite numerous attempts using several growth conditions and media, *Sphaerobacter* spp., *Chelatococcus* spp., and *C. thermoautotrophica* StC could not be isolated in either a nutrient-rich or nutrient-depleted growth medium.

The thermophilic, autotrophic, and aerobic bacterial con-sortium CNF has been growing for 3 years under harsh condi-tions (nutrient-poor environment with a trace gas atmosphere), strongly suggesting the occurrence of carbon and nitrogen fixa-tion. Collectively, the metagenomic data suggest that the CNF consortium can scavenge atmospheric $H_2$, $CO_2$, and CO for energy and carbon sources and that *C. thermoautotrophica* StC is primarily responsible for providing carbon into the consortium biomass through $CO_2$ fixation. Moreover, *C. thermoautotrophica* StC, together with *Chelatococcus* spp. and *Sphaerobacter* spp., possesses the ability to obtain and/or conserve energy via oxi-dation of atmospheric trace gases ($H_2$ and/or CO). Consistent with these findings, pure culture experiments have demonstrated that scavenging of trace gas enables the persistence of diverse heterotrophic aerobes under conditions of organic carbon starvation[34].

Positive interactions are typically considered rare. Nonetheless, the prevalence of positive interactions and the conditions under which they occur are not clearly understood. Kehe et al.[35] attempted to address this topic using kChip, an ultrahigh-throughput coculture platform, to measure 180,408 interactions among 20 soil bacteria across 40 carbon environments. They observed that positive interactions, often described as rare, occurred commonly and primarily as parasitism between strains possessing different carbon consumption profiles.

Altogether, data obtained in the present study suggested that *C. thermoautotrophica* was a keystone species in this model eco-system. The four-species consortium in this study, representing the extreme oligotrophic ecosystem, may be useful for investi-gating bacterial interspecies interactions and microbiome assembly under such extreme conditions. Although carbon fixa-tion could be demonstrated and the keystone species in this consortium could be identified, the consortium housed an intrinsic but unclear nitrogen assimilation mechanism. The ability of these microorganisms to grow and thrive in this med-ium using nitrogen from air suggested the existence of a highly efficient strategy to exploit this nutrient.

Overall, the microbial community structure of the CNF was considered to be shaped by selection for bacteria that could persist under these physically extreme, chemically deprived growth conditions. Therefore, we propose the breadwinner hypothesis to explain the functioning and interactions of this model consortium for microbial cooperation and carbon fixation under strict oligotrophic conditions. This term, which is used in social sciences, refers to a person who earns money to support their family[36]. In our consortium, *C. thermoautotrophica* StC is the breadwinner that is important for the survival of all con-sortium members and the only member capable of remaining metabolically active under these growth conditions. *C. thermo-autotrophica* StC is the sole member that exhibited genes for a known carbon fixation pathway. In this regard, *C. thermo-autotrophica* StC fixes carbon, thereby allowing the incorporation

of this element into the system, which is essential for the survival of other members.

The loss of *C. thermoautotrophica* StC would be fatal to the other members of this consortium. The other members would be the so-called beneficiaries, where *Chelatococcus* spp. and *Sphaerobacter* spp. would be the collaborators (members that are non-essential but provide some benefits to the group), and *Geobacillus* spp. would be an opportunist or cheater, at least considering the available data and energy requirements.

Further studies using gene expression assays are required to compare gene expression and metabolomics in both the absence and presence of nitrogen to help elucidate the mechanisms underlying the survival of this consortium. Despite the evidence regarding incomplete nitrogen assimilation pathways, we cannot rule out the possibility of the existence of other breadwinners in the consortium. The nitrogen assimilation mechanism is still unclear and could be a key component associated with specific members of the consortium. Therefore, the isolation of other members of the consortium, in addition to *Geobacillus* sp. (for example, using specific antibiotics and culture conditions, based on the data obtained from the MAGs) may be useful to clarify key mechanism(s) via in-vitro assays; compared with predictions using metagenomic data, such assays are more stable and facilitate the study of biochemical functions with greater accuracy. For example, several apoenzymes involved in carbon and nitrogen metabolism exhibit post-translational modifications and cofactor maturation (such as molybdopterin and Fe-heme)[37]. Isolation of the main component, *C. thermoautotrophica* StC, would also allow us to examine, model, manipulate, and design metabolic pathways in natural and constructed microbial systems using systemic approaches and synthetic biology. However, we can state that all beneficiaries are completely dependent on the energy provided by the breadwinner *C. thermoautotrophica* StC, whereas the input from other members could contribute to the efficiency of the consortium's survival in such a nutrient-depleted medium.

Our data provide insights into some of the fundamental topics in environmental microbiology, such as microbial cultivability[38,39], and should be useful to support other studies exploring ecological interactions and biotechnological applications of microbial extremophiles.

## Methods

**Soil sampling and characterization**. Overall, five samples were collected from loamy sand soil (0–5-cm layer) under a pile of burned grass with a long history of burning in the municipality of Seropédica, Rio de Janeiro, Brazil (22°46′34.59″ S 43°41′30.71″ W). The soil sample collected from the surface (0–5 cm) was presumed to have a greater exposure to heat and higher concentrations of gases, including CO, $CO_2$, $H_2$, nitrogen oxide (NOx), nitrous oxide ($N_2O$), and $NH_3$[40], owing to vegetation burning[27]. Soil was collected using spatulas disinfected with 70% ethanol. As required by the Brazilian legislation on access to biodiversity (Law 13,123/15 and Decree 8.772/16), we obtained proper authorization from SisGen (National System for the Management of Genetic Heritage and Associated Traditional Knowledge; collection license number: AC72B83). The sampling site was an open field where organic plant residues, such as tree cuttings and grass, were routinely discarded and burned for >15 years (Supplementary Fig. 1A, B). The soil samples were immediately transported to the Molecular Microbial Ecology Laboratory (LEMM), Federal University of Rio de Janeiro (UFRJ), Rio de Janeiro, Brazil. A control soil sample was collected from a nearby site with no history of burning. The physical and chemical properties of all soil samples were determined in triplicate, using standard laboratory protocols[41,42].

**Selection of the CNF consortium**. Aliquots of 0.5 g of each soil sample were inoculated in 100-mL vials containing 40 mL of mineral medium for autotrophs[43] supplemented with a trace element, a vitamin solution, and 4 mM $NH_4Cl$[44]. The $NH_4Cl$ was added only for the initial round of enrichment. The soil-inoculated medium was incubated at 55 °C until microbial growth was observed. Following the appearance of microbial growth in the medium, the colonies, observed as white floating pellicles, were transferred to another flask containing 40 mL of a nitrogen-free medium (N-FIX)[45]. To prepare N-FIX, solution A containing 0.3 g $MgSO_4$•$7H_2O$, 0.2 g NaCl, 0.1 g $CaCl_2$•$2H_2O$, and 12.6 mg $Na_2MoO_4$•$2H_2O$, supplemented with 2 mL of a trace element solution (100 mg/L $ZnSO_4$•$7H_2O$,

30 mg/L $MnCl_2$•$4H_2O$, 300 mg/L $H_3BO_3$, 200 mg/L $CoCl_2$•$6H_2O$, 10 mg/L $CuCl_2$•$2H_2O$, 20 mg/L $NiCl_2$•$6H_2O$, 900 mg/L $Na_2MoO_4$•$2H_2O$, and 20 mg/L $Na_2SeO_3$)[44], and solution B, which was a buffer solution (0.9 g/L $K_2HPO_4$ + 0.1 g/L $KH_2PO_4$), were mixed in a proportion of 3:1, respectively. The medium was adjusted to pH 7.2. Cultures were maintained in the N-FIX medium, and floating pellicles were individually transferred to vials containing 40 mL of fresh N-FIX medium using a platinum loop in a laminar flow cabinet. The 100-mL vials (containing 40 mL of the medium) were hermetically sealed using a rubber cap and aluminum seal, and the headspace was filled with synthetic air (80% $N_2$ + 20% $O_2$), CO, and $CO_2$ at a ratio of 50/45/5. The gas solution applied was previously collected from canisters, using a sterile syringe, and filtered in Millipore syringe filters (0.2 μm). Sterile needles were used throughout the process. Samples were then incubated statically at 55 °C for 1 month[46]. Non-inoculated vials were kept as a negative control for all inoculations and transfers.

The colonies died or sporulated after > 1 month of incubation; moreover, we were unable to revive glycerol stocks after freezing at temperatures of −80 °C and −20 °C. The cell cultures in the N-FIX medium were maintained in an oven at 55 °C and reinoculated with fresh medium every 2 weeks. Of the original soil samples, microbial growth was observed in only three flasks containing the mineral soil medium for autotrophs[43] and in only one flask containing the N-FIX medium. The cultures showing growth were then individually replicated in several flasks, and at least three replicates were kept alive for the duration of the experiment. The control soil samples showed no microbial growth during the same incubation period. For subsequent analyses, the replicates were combined to improve biomass recovery.

**Assessment of microbial growth and tentative isolation of consortium members**. To determine the ability of CNF to grow under highly restrictive conditions with no trace of contaminating ammonium, the N-FIX medium was solidified using 1.5% high-purity Noble agar (Sigma-Aldrich, St. Louis, MO, USA) and modified with 0.3% clinoptilolite (an ammonia-scavenging zeolite)[47,48]. To isolate the consortium members, a set of different culture media, consisting of solidified N-FIX[45], Reasoner's 2 A agar (R2A) (DSMZ medium 830), trypticase soy agar (DSMZ medium 1617), Luria–Bertani broth (DSMZ medium 381), JNFb broth[1], or super optimal broth with catabolite repression (SOC)[49] were used with incubation at different temperatures (28 °C, 37 °C, or 55 °C) for 15 days.

**Microscopic analyses**. A phase-contrast microscope (Zeiss AxioPlan 2) was used to monitor the culture growth at 3, 7, and 15 days post-inoculation. Aliquots of uninoculated N-FIX medium were analyzed for the presence of any cells. An FEI Morgagni microscope (Thermo Fisher Scientific) was used for transmission electron microscopy (TEM) and an FEI Quanta 250 scanning electron microscope (FEI, The Netherlands) was used for scanning electron microscopy (SEM) at the National Center for Structural Biology and Bioimaging, UFRJ. Samples were washed at least twice in 0.01 M phosphate-buffered saline (PBS, pH 7.2) to remove all non-adhered bacteria from the filamentous growth, and then fixed in 2.5% glutaraldehyde and 4% formaldehyde in 0.1 M cacodylate buffer for 1 h at room temperature (approximately 22 °C). Subsequently, the samples were washed in the same buffer, post-fixed in 1% osmium tetroxide ($OsO_4$) and 1.25% potassium ferrocyanide for 2 h, and then dehydrated in a series of increasing ethanol concentrations (30%, 50%, 70%, 90%, 100%, and ultra-dry ethanol) for 30 min at each dehydration step. For SEM, samples were critical point-dried in $CO_2$, coated with gold, and observed with the FEI Quanta 250 microscope. For TEM, dehydrated samples were slowly embedded in Spurr resin (Electron Microscopy Sciences, USA). Ultrathin sections of the samples were stained with uranyl acetate and lead citrate, and images were obtained using an FEI Spirit transmission electron microscope. All images were processed using Adobe Photoshop CS5 software (Adobe Systems Co., USA). The Fast Fourier Transform (FFT), which is a mathematical representation of the spatial frequencies of the image[25], and measurements (Supplementary data 1) were made with the Digital Micrograph software (Gatan, Inc.)

**Microbial community DNA extraction**. For community analysis (amplicon metabarcoding and Denaturing Gradient Gel Electrophoresis, DGGE), total DNA from both the original consortia and the consortia obtained after specific time points following serial transfers in N-FIX medium was analyzed using an FastDNA™ SPIN Kit for Soil (MP Biomedicals) per the standard protocol. For metagenomic analysis, a consortium sample obtained after 10 transfers via the N-FIX medium was used for DNA extraction. To improve cell recovery, 1 mL of 10% (volume/mass of approximately 0.03 M) sodium dodecyl sulfate solution was added to each flask containing culture medium, with a final concentration of approximately 0.25% (0.0008 M). The contents of three glass bottles (120 mL) containing the full-grown consortia were filtered through sterile 0.22-μm Millipore membranes; the membranes were then directly subjected to DNA extraction, and instead of the single lysis step in the FastPrep™ system (Bio 101, Inc., La Jolla, CA, USA), we performed the lysis step again. We used the FastPrep™ system at 6.0 rpm for 40 s. Nanodrop™ and Qubit™ (Thermo Fisher Scientific) were used to assess the quality (260/230 and 260/280 ratios) and amount of the extracted DNA, respectively. Electrophoresis was performed using 1.0% agarose gel for 40 min at 90 V,

after which the gels were stained with SYBR™ Safe (Thermo Fisher Scientific) and observed under an ultraviolet (UV) transilluminator to assess DNA integrity.

**Polymerase chain reaction (PCR) detection of *nifH*.** The presence of nitrogen-fixation systems in the consortia was determined using PCR by targeting the *nifH*, a gene that encodes the dinitrogenase reductase subunit of the nitrogenase enzyme, which is a biological marker for nitrogen fixation[50]. The primers PolF and PolR[51] were used according to the following protocol: initial denaturation at 94 °C for 1 min, followed by 30 cycles of denaturation at 94 °C for 1 min, annealing at 55 °C for 1 min, extension at 72 °C for 2 min, and final amplification at 72 °C for 5 min. *Herbaspirillum seropedicae* ATCC 35892 and *Escherichia coli* BL21 were used as positive and negative controls in JNFb and SOC media, respectively[1]. Electrophoresis of the amplicons was performed on 1.5% gels for 40 min at 100 V, after which the amplicons were stained with SYBR Safe and observed under a UV transilluminator.

**Consortium stability via denaturing gradient gel electrophoresis (DGGE) and amplicon sequencing.** PCR amplification of the extracted DNA was performed using the primers U968-gc (5′-CGCCCGGGGCGCGCCCCGGGCGGGGCGGGG GCACGGGGGAACGCGAAGAACCTTAC-3′) and L1401 (5′-CGGTGTGTACAA GGCCCGGGAACG −3′). A hypervariable region of the gene *rrs* was amplified, and gene codification of the ribosomal RNA composing the small subunit of the bacterial ribosome (16 S) was performed[52]. The amplification reaction was performed as described by Machado de Oliveira et al.[53]. The results of the amplification reaction were determined via electrophoresis of samples on 1.5% agarose gel at 100 V for 40 min. The gel was stained with SYBR Safe and observed under a UV light transilluminator. DGGE was performed using the DCode™ System (Bio-Rad Laboratories, Inc., Hercules, CA, USA) at 70 V and 65 °C for 16 h. The denaturation gradient gel was prepared using a peristaltic pump and a gradient generator block. The gradient of the denaturing agents, urea and formamide, was in the range of 35–65%, and the gel contained 6% polyacrylamide. Gels were stained with SYBR Safe for 20 min and observed using a Storm® gel scanner (General Electric). Before the PCR products were applied to the DGGE gel, they were quantified and normalized to ensure that any changes in band intensities reflected the changes in their relative abundances.

The DNA obtained from the serial transfer in N-FIX was also used to amplify the hypervariable regions V5 and V6 of the bacterial 16 S rRNA gene (according to ref. [16]), at the Argonne National Laboratory (http://ngs.igsb.anl.gov, Lemont, IL, USA) through the Next Generation Sequencing Core on an Illumina MiSeq System (Illumina, San Diego, CA, USA), following the manufacturer's guidelines. The microbial data were also analyzed according to Santoro et al.[16]. The data have been deposited with links to BioProject accession number PRJNA373874 in the DDBJ BioProject database: along with the accession numbers SAMN33200710, SAMN33200711, SAMN33200712, SAMN33200713, SAMN33200714; submission: SUB12816063.

**Metagenomic sequencing, assembly, binning, and data analysis.** Paired-end sequencing was performed using forward and reverse primers complementary to the Nextera™ DNA Library Preparation kit adapters (Illumina®) at MR DNA Lab, Shallowater, Texas, USA (http://www.mrdnalab.com/). To remove low-quality metagenomic reads, the reads were trimmed using the Sickle software (version 1.33)[54]. Subsequently, they were taxonomically classified on CDS using the Kaiju taxonomic classifier (version 1.7.3)[55]. As a reference database, the nonredundant National Center for Biotechnology Information (NCBI) protein sequences (accessed June 25, 2021, containing protein sequences from bacteria, archaea, and viruses) were used. Raw reads of the consortium sample were deposited in the NCBI Sequence Read Archive under accession number SRR12323727. The *Carbonactinospora thermoautotrophica* MAG contigs were deposited in the NCBI under the WGS master record PQID00000000.1 and Biosample SAMN07787832.

The reads belonging to *rrs* (16 S rRNA gene) sequences were obtained from all metagenomic contigs, using the Barrnap ribosomal predictor software (https://github.com/tseemann/barrnap). To reconstruct the phylogeny of the filtered *rrs* sequences, we used the Seq Match tool from the Ribosomal Database Project database (*rrs* –only sequences of >1200 bp derived from culturable microorganisms). The five most similar matches were selected for each contig and gene and aligned using the MUltiple Sequence Comparison by Log-Expectation software (version 3.8.31). Phylogenic reconstruction was performed with MEGA-7[56], using the maximum likelihood method and the Tamura–Nei distance model[57] with 1000 bootstrap replications.

**Metagenome-assembled genomes (MAGs) and genomic comparison.** Metagenomic sequencing reads were assembled using the SPAdes software (version 3.8.0)[58]. Contigs belonging to each consortium member were grouped using the CAR contig assembly tool[59] with complete reference genomes from the NCBI database to obtain the MAGs. The BLAST Ring Image Generator (BRIG) software (version 0.95-dev.0004)[60] was used to display circular comparisons between the MAGs obtained from the consortium and those from the NCBI reference genomes. Alignments were generated by BRIG using the BLASTN program (version 2.4.0+)

and the concatenated contigs of MAGs (from the present study) or the previously isolated LEMMY01[23] and reference strain sequences (UBT1, H1, DMS18167, DSM20745) and metagenomic reads. Additionally, the DNAPlotter software (version 17)[61] was used to highlight the GC content plot.

**Nitrogen cycle.** To identify potential genes encoding nitrogenases and to complement the *nifH* PCR results, including the classical and alternative enzymes previously described, the reads associated with the nitrogen metabolic pathway were analyzed in MEtaGenome Analyzer 5 (MEGAN5) software[26], using the Kyoto Encyclopedia of Genes and Genomes (KEGG) Pathway and NCBI databases as references.

**Metabolic complementation analysis.** Overlapping paired-end reads were merged using the Paired-End reAd merger software[62]. Merged and unmerged reads were mapped to MAGs using Bowtie2 software[63] to extract the reads belonging to each consortium member, including the missing genes in the reference genomes. The BlastX tool was used to align the extracted reads to the NCBI nr database. Metabolic reconstructions for the consortium members were obtained using the KEGG Pathway database and MEGAN5 software[26]. The differences in metabolic pathways, heteromultimeric enzymes, and transporters were highlighted to assess the possibility of metabolic complementation among consortium members. The CheckM software was used to evaluate the completeness of MAGs[64]; the complementation between the main metabolic pathways was assessed by submitting the MAGs and published whole-genome sequencing (WGS) data of *Geobacillus* LEMMY01[23] into the metabolisHMM tool[65].

**Isotope ratio mass spectrometry (IRMS)-labeled nitrogen assay.** Experiments were performed using $^{15}N_2$-labeled atmospheric nitrogen (98% $^{15}N$ atoms; Cambridge Isotopes, Tewksbury, MA, USA). To evaluate nitrogen fixation, the consortium cells were grown in 40 mL of liquid N-FIX medium with or without 1.5 g/L (28 mM) $NH_4Cl$ and solidified with 0.6% gellan gum in 100-mL flasks. *H. seropedicae* ATCC 35892 and *E. coli* BL21 were used as positive and negative controls, respectively, in JNFb and SOC media[1]. Gaseous atmosphere with the addition of 10 or 30 mL of $^{15}N_2$ was typically used for conducting the transfers of consortium cells. For *H. seropedicae* and *E. coli*, 10 mL of $^{15}N_2$ + 2% $O_2$ was added. Samples were incubated for 5 and 14 days at 55 °C (CNF) or 30 °C (*H. seropedicae* and *E. coli*). Following incubation, any growth present was scraped from the vials and washed using PBS. The washed samples were dried at 60 °C for 24 h. Next, a Denver Instruments M-220D digital balance was used to obtain samples weighing 0.5–2 mg; they were then placed in capsules, inserted into the wells of a 96-well microtiter plate, and sealed using strip caps. The $^{15}N/^{14}N$ isotope ratio was determined using IRMS (University of California Davis, Stable Isotope Facility, Davis, CA, USA).

**Statistics and reproducibility.** The physical and chemical properties of all soil samples were determined in triplicate. The bacterial cultures were replicated in several flasks at multiple times, and at least three replicates were kept alive during the duration of the experiment. The stability of the consortium was also monitored over three years. Control soil samples and non-inoculated culture media showed no microbial growth during the same incubation period. Merged and unmerged reads were mapped to MAGs using Bowtie2 software[63] and BlastX was used to align the extracted reads to the NCBI nr database. Metabolic reconstructions were performed using the KEGG Pathway database and MEGAN5 software[26] and the complementation between the main metabolic pathways was assessed through metabolisHMM[65].

**Reporting summary.** Further information on research design is available in the Nature Portfolio Reporting Summary linked to this article.

## Data availability
All data supporting the findings of this study are available within the article and its Supplementary Information files. Amplicon sequencing can be accessed on DDBJ BioProject ID PRJNA373874: along with the accession numbers SAMN33200710, SAMN33200711, SAMN33200712, SAMN33200713, SAMN33200714; submission: SUB12816063. The raw metagenome sequencing data was deposited in the NCBI Sequence Read Archive under accession number SRR12323727. The *Carbonactinospora thermoautotrophica* MAG contigs were deposited in the NCBI under the WGS master record PQID00000000.1 and Biosample SAMN07787832. The source data underlying Figs. 2c and 5 are provided as Supplementary Data 1–2, respectively. Additional information and relevant data are available from the corresponding author upon reasonable request.

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

## Acknowledgements

This research was financially supported by the National Council for Research and Development (CNPq), the National Council for the Improvement of Higher Education (CAPES), and a KAUST Baseline Grant (to Prof. A. S. Rosado) (BAS/1/1096-01-01). We thank Edir Martins Ferreira and Tahira Jamil for their excellent technical assistance and Prof. E. Zonta for soil analysis. We would like to remember Professor Ulysses Lins, who is sadly no longer with us and to whom we owe much more than words.

## Author contributions

A.S.R. conceived and designed the study; Y.P., U.L. and F.M. performed the experiments; Y.P., F.M., R.S.P, J.D.V., J.L.M.R., and A.S.R. analyzed the data; all authors discussed the results and contributed to the manuscript preparation. A.S.R. and J.L.M.R. provided financial support. All authors read and approved the final manuscript.

## Competing interests

The authors declare no competing interests.
