## [Peer Review File · Communications Biology]

Reviewers' comments:

Reviewer #1 (Remarks to the Author):

In their manuscript the authors try to demonstrate the metabolic linkages between the 4 members of what they refer to as a Carbonitroflex community. Based on the metagenome sequence data obtained they postulate that the most dominant member of the community, linked to the genus *Carbonactinospora*, is the keystone or "breadwinner" member that provides the carbon via CO₂ fixation and could be involved in nitrogen assimilation.

Although the manuscript provides some interesting data and hypotheses on the possible interactions between members of the community, the manuscript needs editing to enable the reader to easily follow the experiments and the subsequent analyses and findings. The authors could even consider splitting the manuscript into two, one dealing with the development of the community and the identification of the main members and another focusing on the metabolic linkages between the community members. I am not sure that the data presented in this paper fully answers Question one: "How did the community respond to these growth conditions over time?"

There are several issues that are not clearly captured in the methods section. No information is provided on the procedure followed during the 10 transfers. Were there multiple vials per transfer? Were they treated as separate lines during the transfers or were they mixed before each transfer? Were the four samples which were combined (4 X 40 ml = 160 ml but you mentioned 100 ml?) all belonging to the 10th transfer?

In the methods section the authors report the inclusion of a control sample, but apart from the soil analyses nothing else is reported. Was there no growth in the control? Was the composition of the test community analyzed at time zero? This will help the reader understand if the test community had already been exposed to selective conditions favoring the development of the CNF community.

The identification of the different MAGs based on the 16 S rRNA gene could be improved by including the sequences of the type strains of the potential species. The presentation of Figure 2 should be improved. Some of the labels of the taxa are incomplete and one has to guess that the sequences marked with a diamond are the sequences retrieved from the MAGs. The labels used in the Extended Data Figure 3 are even more confusing. The 16 S sequence of the *Geobacillus* MAG should also be included to show that LEMMY01 is a true representative of that community. It would also be ideal to include recruitments plots to show how well LEMMY01 represents the community.

The issue of the highly divergent 16 sequences associated with the StC Mag could be unpacked in more detail by including some of those divergent sequences from the previous studies such as Volpiano et al. 2021.

What are the differences between figure 3 A and C?

With the limited information on the interactions between the different members of the consortium it may be premature to label the *Geobacillus* population as cheaters.

Minor comments:

Line 33: You are not describing a new species so the use of the words "Candidatus strain" is confusing.

Line 85: Spelling of carboxydrotrophs

Line 92: The consortium was not isolated from the soil sample; the soil sample was used for the enrichment of the consortium.

Line 163: Rather use filamentous growth instead of hyphae

Line 442: Do not refer to StC as a strain as it is a MAG

Reviewer #2 (Remarks to the Author):

The manuscript titled "The Breadwinner Hypothesis: Collaborators and Opportunists in a Thermophilic Chemolithoautotrophic Bacterial Consortium (Carbonitroflex)" by Pinheiro et al., describes the characterization of a microbial consortium able to grow in poor nutrient medium. Authors successfully isolated the consortia from soil samples and were able to identify each of the member by microscopy and metagenomics analysis. Based on the date, authors describe the individual metabolic capacities for each member and describe the possible interactions between them proving some insights in the use of resources in a poor nutrient environment. The bread-winner hypothesis was proposed to better describe the interactions in the consortia. The study shows that microbial consortia are able to survive in harsh environments by establishing different relationships.

Major comments

Despite the authors managed to obtain valuable data derived from the characterization of the community, the organization and presentation of the data confusing. The individual sections are very short and provide little information of the findings of that particular experiment, which at the beginning leaves a lot of questions and gaps. These gaps are latter addressed in the long discussion. Suggesting rearranging the information to enhance continuity in the text, so the overall readability of the manuscript is improved. Additionally, the overall format and organization of the figures is not consistent, causing difficulty to locate the data.

Authors propose the idea of the breadwinner hypothesis; they propose that the only member able to fix carbon is Carbonactinospora, this is reasonable if carbon is considered as the only limiting factor, but since nitrogen assimilation mechanism in the community still unclear, nitrogen could also be playing the role of "the bread" thus the possibility of multiple breadwinners. Could the authors provide more insights about this idea?

Authors mentioned that the only member of the consortia which was successfully isolated was Geobacillus. What would be explanation for this? is this suggesting that this member could be providing something essential for the other members? Please clarify.

Authors mention the existence of carboxisome-like structures, but few information and discussion is provided about them, since the carbon fixation is critical for the community. the carbon fixation is spatial?? Please expand this idea.

Suggest moving parts of the discussion to expand the nitrogen fixation section. Since N assimilation is also a critical survival of the community and the exact mechanism still unknown, suggest adding a new figure or panel describing a possible N uptake mechanism.

Minor comments

Figure 1 Legend for each panel cannot be seen properly, suggesting moving it outside the picture for a better reading. Additionally, panels D-E show many of the mentioned well-defined structures in the cells, vacuole, carboxysomes, however none of these structures are pointed of highlighted in the photos, which makes difficult for the reader to identify these structures. D

Figure 5, is not clear, the way panels and division are organized, gives the idea that Rubisco and carbon fixation is performed by both sphaerobacter and chelatococcus, it's also not clear what would be Geobacter exchanging with the other members. The final fate of the energy produced in the membrane is not indicated. This energy is only produced by carbonactispora? Reorganizing and adding the missing information to the figure can improve it.

Reviewer #3 (Remarks to the Author):

This is a fairly good manuscript describing the isolation of a stable temophilic carboxydrotrophic community that develops on a mineral medium under conditions of nutrient limitation, as well as a probable scheme of interaction within this community.

The next section includes a number of critiques, some optional and some what I consider required

suggestions for improvement.

The Abstract well describes the context of the work and the main results. However, there are some inconsistencies in the text. The authors declare that interactions between all members of the community are necessary for the survival of the group under cultivation conditions because no pure cultures were obtained. At the same time, it is argued that only *Carbonactinospora thermoautotrophica* are vital for the survival of all consortium. Also, the work does not demonstrate that the exclusion of one of the secondary members of the community leads to the impossibility of its growth under cultivation conditions.

The Introduction is well-written, with clear objectives for the study, but I suppose one or two short examples of specific "novel biotechnological applications" or "novel bioproducts" might be a good addition.

The Methods and Results are generally very detailed and descriptive (see exceptions below).

Line 115, 142 Need to clarify - were the samples taken aseptically and the gases filtered?

Line 125 The cited reference (Embrapa, 1997) is not listed in References

Line 133 Here, the abbreviation of the medium is given as NFIX, while the spelling N-FIX is also implied later in the text.

Line 134-138 Please check if the rules of the journal allow the indication of concentrations in the form g/L, mg/L (I suppose unit dimensions should be expressed using negative integers). $\text{CaCl}_2 \cdot \text{H}_2\text{O}$ is it $\text{CaCl}_2 \cdot 2\text{H}_2\text{O}$? What was the pH of the medium?

Line 180 The final concentration of SDS in the lysis buffer should be indicated.

Line 190 Here it is necessary to give a decryption of nifH

Line 198-212 Were the sequences resulting from the DGGE deposited in the NSBI database or others?

Line 202 Here it is necessary to give a decryption of rrs.

Line 316-321 Why was the tactic of culturing the community on different media but not antibiotic treatment chosen, taking into account the presence of both Gram-positive and Gram-negative bacteria in the consortium?

Line 339-341 Here the authors indicate that "(8% of reads) were related to Chelativorans" however, the rest of the text refers to a member of the genus *Chelatococcus* It here and in Fig2 is also necessary to give the correct names of the phyla.

Extended Data Figure 1 The table needs careful editing

Discussion

In general, the authors presented a good discussion of the results obtained using modern methods. It is obvious that additional experimental studies are required to elucidate all aspects of the functioning of the described community. In particular, both the omics approach and chemical analysis of the environment in order to identify metabolites exchanged among community members, as well as possible sources of nitrogen. It would also be interesting to know how significant the role of such a consortium is in the natural community of the studied soil. I hope this will become a task for future work.

Reviewers' comments:

Reviewer #1 (Remarks to the Author):

In their manuscript the authors try to demonstrate the metabolic linkages between the 4 members of what they refer to as a Carbonitroflex community. Based on the metagenome sequence data obtained they postulate that the most dominant member of the community, linked to the genus Carbonactinospora, is the keystone or “breadwinner” member that provides the carbon via CO₂ fixation and could be involved in nitrogen assimilation.

Although the manuscript provides some interesting data and hypotheses on the possible interactions between members of the community, the manuscript needs editing to enable the reader to easily follow the experiments and the subsequent analyses and findings. The authors could even consider splitting the manuscript into two, one dealing with the development of the community and the identification of the main members and another focusing on the metabolic linkages between the community members. I am not sure that the data presented in this paper fully answers Question one: “How did the community respond to these growth conditions over time?”

A: Thank you for your comments and feedback. We were delighted to see the interest expressed in this work and the critical feedback provided, even suggesting the paper to be split. We believe however, that the story is more robust and interesting if we include all of the data in the same manuscript, as these mechanisms are all connected. We have therefore, based on the comments (and detailed below) made major changes to our manuscript, including a full restructuring of the text and figures, as well as the incorporation of new data and images. We have also edited the wording to better reflect the results, in order to keep the full story into the same paper, but in a much clearer way.

There are several issues that are not clearly captured in the methods section. No information is provided on the procedure followed during the 10 transfers. Were there multiple vials per transfer? Were they treated as separate lines during the transfers or were they mixed before each transfer? Were the four samples which were combined (4 X 40 ml = 160 ml but you mentioned 100 ml?) all belonging to the 10th transfer?

A: We agree this was not clear and included information to better explain the process, from the beginning.

First, regarding the original samples, we have added some information to clarify the procedure as follows:

Lines 128-134: “Among the original soil samples, microbial growth was observed in only three flasks containing the mineral soil medium for autotrophs (Meyer & Schlegel, 1978) and in only one flask containing the N-FIX media. The cultures

showing growth were then individually replicated in several flasks, and at least three replicates were kept alive during the duration of the experiment. The control soil samples showed no microbial growth during the same incubation period. For subsequent analyses, the replicates were combined to improve biomass recovery.”

In addition, we have clarified the volumes. The first mentioning of 100ml was referring to the flask volume, not the content. Each flask contained 40ml of medium, and three of these flasks were combined for DNA extraction (total of 120mL). This information was added, as detailed above and as follows:

Lines 115-123: “The medium was adjusted to pH 7.2. Cultures were maintained in the N-FIX medium, and floating pellicles were individually transferred to vials containing 40 mL of fresh N-FIX medium using a platinum loop in a laminar flow cabinet. The 100-mL vials (containing 40mL of the medium) were hermetically sealed using a rubber cap and aluminum seal, and the headspace was filled with synthetic air (80% N₂ + 20% O₂), CO, and CO₂ at a ratio of 50/45/5 using a sterile syringe and Millipore syringe filters (0.2 µm). Samples were then incubated statically at 55°C for 1 month (Gadkari et al., 1990).”

In the methods section the authors report the inclusion of a control sample, but apart from the soil analyses nothing else is reported. Was there no growth in the control? Was the composition of the test community analysed at time zero? This will help the reader understand if the test community had already been exposed to selective conditions favouring the development of the CNF community.

A: Thank you for pointing this out, this is indeed a crucial information that was missing. The control sample consisted of a soil sample from a nearby site without soil burn history. We observed no growth in the control samples (negative controls over the passages). This information was added to the text (e.g., methods: “Non-inoculated vials were kept as negative control for all inoculations and transfers.” and in results: “The consortium was able to grow and form floating white pellicles in a mineral medium without any source of organic carbon and nitrogen, and no growth was observed for any of the negative controls applied.”).

As for the test community, we believe the constant pressure of temperature and CO₂ for vegetal burn enriched the thermophilic consortia in the soil, but that the culture conditions have also selected an even more specific community, as the original community that was able to grow under the applied conditions is slightly different from the community observed over the years. We have included new Illumina sequencing data from the original consortium, and the community observed over 10 transfers (up to three years). We added data to show these details, as follows (and in Supplementary Figure 2):

Lines 329-336: “The consistent stability of the consortium was evaluated using DGGE community profiling and 16S ribosomal RNA gene barcoding over a period of approximately 3 years of cultivation (44 transfers) at variable intervals of months (Extended Data Fig. 2A and B). During this period, the DNA of the consortium was extracted at different times (after 5, 12, 16, 36, and 44 transfers), and DGGE and

Illumina sequencing were performed to compare the profile of the bacterial community based on the ribosomal gene *rrs*. The results suggested that despite small fluctuations and changes in band intensities (even after normalization), and bacterial composition, the bacterial core of the consortium remained stable (Supplementary Fig. 2A and B).”

Supplementary Figure 2. Denaturing gradient gel electrophoresis (6% acrylamide gel containing a urea and formamide gradient [range: 35%–65%] run at 70 V for 16 h) of *zcz* gene fragments from samples collected from the consortium over time. Transfers occur each 15 days. I) After 5 transfers; II) after 12 transfers; III) after 16 transfers; IV) after 36 transfers; and V) after 44 transfers (approximately 3 years). (A). Bar plot showing the taxonomic composition of the *carbonitroflex* consortium obtained via *zcz* gene amplicon sequencing (Illumina) in the year of its isolation (1) and after consecutive transfers (i.e., after 1 year (2), 2 years (3) and 3 years (4) (B).

The identification of the different MAGs based on the 16 S rRNA gene could be improved by including the sequences of the type strains of the potential species.

A: We agree, the type strain has been included. We used SILVA database to search for the 5 closest hits (type strains) for each of the 16S sequence. We believe this will accurately predict the taxonomy, alongside with the taxonomic binning.

The presentation of Figure 2 should be improved. Some of the labels of the taxa are incomplete and one has to guess that the sequences marked with a diamond are the sequences retrieved from the MAGs.

A: Figure 2 (which is now Fig. 3, as new data was provided as Fig. 2) has been fully revised. A new pie chart and a new phylogenetic tree were done, with improved visualization. New labels for the 16s sequences retrieved from the MAG's are now stated as part of the legend as marked with the black diamond shape.

Figure 3

The labels used in the Extended Data Figure 3 are even more confusing. The 16 S sequence of the Geobacillus MAG should also be included to show that LEMMY01 is a true representative of that community.

A: We agree and also decided to delete Supplementary Figure 3, to improve readability since it's a redundant information that is already presented in the new figure 3.

It would also be ideal to included recruitments plots to show how well LEMMY01 represents the community.

A: Thank you for your suggestion, the new version includes a similar analysis. Figure 4 was modified to compare the metagenomic reads for the Geobacillus sp. LEMMY01 genome, in order to show its representativeness within the consortium, using BRIG plot. The representativeness of LEMMY01 (Geobacillus) is also provided in Figure 3.

The issue of the highly divergent 16 sequences associated with the StC Mag could be unpacked in more detail by including some of those divergent sequences from the previous studies such as Volpiano et al. 2021.

A: Done. The 16S sequences from previous studies were included to reconstruct the phylogenetic tree. The reference was included in the text and in figure 3.

What are the differences between figure 3 A and C?

A: Apologies, this was an error, they are the same figure. We replaced the repeated figure by the correct one, containing the Sphaerobacter spp genome. This is currently Figure 4 in the revised text, and it was also overall improved.

Figure 4

With the limited information on the interactions between the different members of the consortium it may be premature to label the Geobacillus population as cheaters.

A: This is a very important discussion indeed. Even not knowing all of the interactions, it is reliable to say that *Geobacillus* sp. is using the energy provided by the breadwinner. Considering its genome, we can not find any known essential contribution that can only be provided by *Geobacillus* sp. Besides, we have isolated *Geobacillus* sp., using different medium and culturing conditions, and tested its ability to grow using the NFIX medium and gas atmosphere (this information was added to the text, as detailed below). As *Geobacillus* sp. cannot grow under this conditions, even if using a high number of cells in the inoculum, it very likely that, under this conditions, *Geobacillus* sp., is a cheater, as it only grows in the presence of others, and that, based on its genome and other metagenomes, it is also very likely that others could grow without *Geobacillus* sp. We have clarified this in the text, adding this caveat and that this idea is based on the available data, energy budget, essential contribution and source of carbon, as follows:

Lines 583-587: “The loss of *C. thermoautotrophica* StC would be fatal to the other members of this consortium. The other members would be the so-called “beneficiaries,” where *Chelatococcus* spp. and *Sphaerobacter* spp. would be the “collaborators” (members that are non-essential but provide some benefits to the group), and *Geobacillus* spp. would be an “opportunist” or “cheater”, at least considering the available data and energetic requirements.

Lines 603- 606: “...However, it is reliable to state that all beneficiaries are completely dependent on the energy provided by the breadwinner *C. thermoautotrophica* StC,

whereas the input from other members could contribute to the efficiency of the consortium's survival in such a nutrient-depleted medium.”

Lines 538-541: “From the consortium, only *Geobacillus* spp. (LEMMY01) could be isolated using a low nutrient Reasoner's 2A medium (R2A), which is one of the most used media for the isolation and growth of oligotrophic, environmental bacteria. However, the same strain was incapable of growing axenically in an N-FIX medium with a gaseous atmosphere.”

Minor comments:

Line 33: You are not describing a new species so the use of the words “Candidatus strain” is confusing.

Line 85: Spelling of carboxydotrophs

Line 92: The consortium was not isolated from the soil sample; the soil sample was used for the enrichment of the consortium.

Line 163: Rather use filamentous growth instead of hyphae

Line 442: Do not refer to StC as a strain as it is a MAG

A: All off these concerns were addressed and the text was edited accordingly. The term “Candidatus strain” were deleted and all other suggestions were incorporated.

Reviewer #2 (Remarks to the Author):

The manuscript titled “The Breadwinner Hypothesis: Collaborators and Opportunists in a Thermophilic Chemolithoautotrophic Bacterial Consortium (Carbonitroflex)” by Pinheiro et al., describes the characterization of a microbial consortium able to grow in poor nutrient medium. Authors successfully isolated the consortia from soil samples and were able to identify each of the member by microscopy and metagenomics analysis. Based on the date, authors describe the individual metabolic capacities for each member and describe the possible interactions between them proving some insights in the use of resources in a poor nutrient environment. The bread- winner hypothesis was proposed to better escribe the interactions in the consortia. The study shows that microbial consortia are able to survive in harsh environments by establishing different relationships.

Major comments

Despite the authors managed to obtain valuable data derived from the characterization of the community, the organization and presentation of the data confusing. The individual sections are very short and provide little information of the findings of that particular experiment, which at the beginning leaves a lot of questions and gaps. These gaps are latter addressed in the long discussion. Suggesting rearranging the information to enhance continuity in the text, so the overall readability of the manuscript is improved. Additionally, the overall format and organization of the figures is not consistent, causing difficulty to locate the data.

A: Thank you for your positive assessment and very productive suggestions. We have fully restructured the text accordingly, to improve flow and continuity. The result section was reorganized and the discussion was shortened into a less convoluted and better organized version. Overall readability was also improved by another round of corrections by the authors and a proofreading service. We have also edited, standardized and replaced and/or improved figures and tables.

Authors propose the idea of the breadwinner hypothesis; they propose that the only member able to fix carbon is Carbonactinospira, this is reasonable if carbon is considered as the only limiting factor, but since nitrogen assimilation mechanism in the community still unclear, nitrogen could also be playing the role of “the bread” thus the possibility of multiple breadwinners. Could the authors provide more insights about this idea?

A: We agree with the reviewer. The nitrogen assimilation mechanism is unclear and may play a key role in interactions as well, and the model is solely based on a metagenomic-based model. Only the future isolation of different members of the consortium could confirm the hypothesis. This is now highlighted in our text as a caveat and subject for further research, as follows:

Lines 588-606: “Further studies using gene expression assays are required to compare gene expression and metabolomics in both the absence and presence of nitrogen to help elucidate the mechanisms underlying the survival of this consortium. Despite the evidence regarding incomplete nitrogen assimilation pathways, we cannot rule out the possibility of the existence of other “breadwinners” in the consortium. The nitrogen assimilation mechanism is still unclear and could be a key component associated with specific members of the consortium. Therefore, the isolation of other members of the consortium may be useful to clarify key mechanism(s) via in vitro assays; compared with prediction using metagenomic data, such assays are more stable and facilitate the study of biochemical functions with greater accuracy. For example, several apoenzymes involved in carbon and nitrogen metabolism exhibit post-translational modifications and cofactor maturation (such as molybdopterin and Fe-heme) (Blake et al. 2022). Isolation of the main component—*C. thermoautotrophica* StC—would also allow us to examine, model, manipulate, and “design” metabolic pathways in natural and constructed microbial systems using systemic approaches and synthetic biology. However, it is reliable to state that all beneficiaries are completely dependent on the energy provided by the breadwinner *C. thermoautotrophica* StC, whereas the input from other members could contribute to the efficiency of the consortium’s survival in such a nutrient-depleted medium.”

Authors mentioned that the only member of the consortia which was successfully isolated was Geobacillus. What would be explanation for this? is this suggesting that this member could be providing something essential for the other members? Please clarify.

A: This data solely indicates that *Geobacillus* is easier to be cultured and isolated, due to limitations in culturing conditions. We are currently working on culture media improvements, based on the metagenomic data, to also provide the necessary conditions that can facilitate culturing the other three members of the consortium. The results will be part of a second paper, more focused on the isolates and on culturing them, not in their interaction. We don't believe this data, alone, would indicate a key role of *Geobacillus* in CNF interactions, but most likely a methodological limitation from our side (overall, as microbiologists).

Authors mention the existence of carboxisome-like structures, but few information and discussion is provided about them, since the carbon fixation is critical for the community. the carbon fixation is spatial?? Please expand this idea.

A: We agree. This idea was expanded (as exemplified below) and new data was also included (Fig 2):

Lines 319-326: "Remarkably, the presence of highly symmetrical bacterial microcompartments (BMCs)/carboxysome-like structures in the cellular cytoplasm were also noted (Fig. 2). Interestingly, the presence of periodic clusters in the cellular cytoplasm could be noted. These diamond-shaped clusters appear in several images and in different sizes (probably due to the block cutting at different heights). Genomic data indicates (see below) that these structures might be carboxysome-like structures, which are bacterial microcompartments that concentrate carbon-fixing enzymes and play essential role in the carbon fixation process (Liu, 2021)."

Lines 381-398: "Being the sole member containing genes supporting CO₂ fixation, *C. thermoautotrophica* StC appeared to be responsible for carbon fixation as it harbored the type I ribulose-1,5-bisphosphate carboxylase (RuBisCO) enzyme, which is a member of the Calvin–Benson–Bassham (CBB) cycle. its shell and accessory enzymes were also found in the StC genome, such as CcmL, annotated as a carbon dioxide concentration mechanism protein. Indeed, structures of bacterial microcompartments (BMC-like) related to carboxysomes were observed by transmission electron microscopy (Fig. 2A-B). The Fourier Fast Transform, which is a mathematical representation of the spatial frequencies of the image (Briegl et al, 2009), showed that the StC carboxysomes present a periodic organization with a well-ordered pattern. The distance between each arrangement is between 7 ~ 8 nm. (Fig. 2C)."

Figure 2

Suggest moving parts of the discussion to expand the nitrogen fixation section. Since N assimilation is also a critical survival of the community and the exact mechanism still unknown, suggest adding a new figure or panel describing a possible N uptake mechanism.

A: Good point. The discussion section has been completely restructured and includes an expanded assessment of the nitrogen fixation aspect. We have also slightly modified Fig. 6 (former Fig. 5) to include the known nitrogen cycle-related genes found in members of the consortium. We have, however, decided not to include a panel suggesting potential mechanisms because, at this point, this would be extremely challenging and speculative - although we have indeed included an extensive discussion about this topic, as for example:

Line 488-536: "...Nevertheless, for long-term survival under highly adverse conditions, such as our N-FIX medium, the consortium may depend on nitrogen fixation and/or a high nitrogen scavenging capacity of one or more members of the consortium who are able to utilize trace amounts of NH_3 . However, in the present study, the consortium including four strains showed no classical nitrogenase-encoding genes. Moreover, we detected no nitrogen fixation activity using the isotope tracer ($^{15}\text{N}_2$) in the experimental setup. No evidence for (classical) N_2 fixation was observed; therefore, the ability of this consortium to grow in the N-FIX medium suggests the existence of an efficient mechanism to obtain the nitrogen for biomass generation.

The possibility that the gaseous mixture was contaminated with trace amounts of NH_4 (despite the 99% purity guaranteed by the company) could not be completely

eliminated. Moreover, trace contamination from air could be present in the medium because NH_3 dissolves extremely rapidly in water.

Yoshida et al. (2014) reported that *Rhodococcus erythropolis* N9T-4, an extreme oligotrophic actinomycete isolate, can grow in minimal medium without a nitrogen source. The nitrogen oligotrophy of N9T-4 involves the strong expression of an ammonium transporter gene (*amtB*); the authors suggested that N9T-4 can utilize a trace amount of atmospheric ammonia as a nitrogen source. Therefore, nitrogen scavenging is another parsimonious explanation for the growth of microbes because it has previously been demonstrated in other microbes surviving in a nitrogen-free medium (Yoshida et al, 2014; Macklear et al., 2016). However, the presence of “super scavenger bacteria” that can thrive under such conditions is unlikely to explain the growth of members of the entire consortium. Moreover, the use of clinoptilolite—a powerful chelator of trace ammonia—and Noble agar did not inhibit the growth of the consortium. Although nitrogen scavenging cannot be completely ruled out because of the hypothetical presence of unknown metabolic paths (for example, the recent description of a novel pathway for BNF described by Higdon et al. 2020), we believe that the strict measures employed in the present study would possibly generate insufficient nitrogen (if any) to support bacterial growth and thus would only encourage the growth of true diazotrophs on the N-FIX medium.

The absence of a classical nitrogen fixation pathway raises questions regarding the strict nitrogen budget in this extremely harsh oligotrophic environment and the need for the residing extremophiles to grow under nitrogen scarcity. One of the primary concerns in the current study was regarding the growth of the consortium without a reduced nitrogen source (NH_3 or organic nitrogen). Theoretically, necromass or viral shunt could sustain some amount of minimal growth (Bradley et al., 2019). In fact, Shoemaker et al. (2021) recently demonstrated that for closed systems, necromass recycling contributed to the maintenance of energy-limited cells as well as facilitated some reproduction. However, explaining the growth of the consortium based only on viral shunt, cannibalism, or necromass use appears highly unreasonable. Therefore, we suggest that the consortium fixed N_2 via an unknown mechanism or that at least one extremely efficient scavenger responsible for the increased biomass for all members was present in the consortium, perhaps providing necromass or organic compounds (including nitrogen sources) to the others.

Furthermore, the presence of proteins involved in nitrification could not be predicted. Moreover, the metagenomes of *Sphaerobacter* spp. and *Chelatococcus* spp. exhibited a lack of crucial genes encoding proteins involved in the final reduction of NO to N_2 ; this resulted in the incomplete mapping of denitrification. However, incomplete denitrification pathways are common in extremophiles, particularly thermophiles (Chen et al., 2002; Hedlund et al., 2011). Typically, denitrification requires low oxygen levels and high nitrate levels, which differ from the growth conditions in the present study.”

and

Lines 587-605: “Further studies using gene expression assays are required to compare gene expression and metabolomics in both the absence and presence of

nitrogen to help elucidate the mechanisms underlying the survival of this consortium. Despite the evidence regarding incomplete nitrogen assimilation pathways, we cannot rule out the possibility of the existence of other “breadwinners” in the consortium. The nitrogen assimilation mechanism is still unclear and could be a key component associated with specific members of the consortium. Therefore, the isolation of other members of the consortium, in addition to *Geobacillus* sp., may be useful to clarify key mechanism(s) via in vitro assays; compared with prediction using metagenomic data, such assays are more stable and facilitate the study of biochemical functions with greater accuracy. For example, several apoenzymes involved in carbon and nitrogen metabolism exhibit post-translational modifications and cofactor maturation (such as molybdopterin and Fe-heme) (Blake et al. 2022). Isolation of the main component—*C. thermoautotrophica* StC—would also allow us to examine, model, manipulate, and “design” metabolic pathways in natural and constructed microbial systems using systemic approaches and synthetic biology. However, it is reliable to state that all beneficiaries are completely dependent on the energy provided by the breadwinner *C. thermoautotrophica* StC, whereas the input from other members could contribute to the efficiency of the consortium’s survival in such a nutrient-depleted medium.”

Minor comments

Figure 1 Legend for each panel cannot be seen properly, suggesting moving it outside the picture for a better reading. Additionally, panels D-E show many of the mentioned well-defined structures in the cells, vacuole, carboxysomes, however none of these structures are pointed or highlighted in the photos, which makes difficult for the reader to identify these structures.

A: Legends have been improved regarding the content, size and background. We have also incorporated the reviewer’s suggestion to indicate the mentioned structures with arrows and different colours.

Figure 5, is not clear, the way panels and division are organized, gives the idea that Rubisco and carbon fixation is performed by both sphaerobacter and chelatococcus, it’s also not clear what would be Geobacter exchanging with the other members. The final fate of the energy produced in the membrane is not indicated. This energy is only produced by carbonactispora? Reorganizing and adding the missing information to the figure can improve it.

A: We apologize, the division was indeed wrong, and we have also found some other typos. Thank you for pointing this out. In addition, we have reorganized other aspects and included the carboxysome and nirB gene that were missing. This is now Figure 6.

Figure 6

Reviewer #3 (Remarks to the Author):

This is a fairly good manuscript describing the isolation of a stable temophilic carboxydrotrophic community that develops on a mineral medium under conditions of nutrient limitation, as well as a probable scheme of interaction within this community.

The next section includes a number of critiques, some optional and some what I consider required suggestions for improvement.

The Abstract well describes the context of the work and the main results. However, there are some inconsistencies in the text. The authors declare that interactions between all members of the community are necessary for the survival of the group under cultivation conditions because no pure cultures were obtained. At the same time, it is argued that only Carbonactinospora thermoautotrophica are vital for the survival of all consortium. Also, the work does not demonstrate that the exclusion of one of the secondary members of the community leads to the impossibility of its growth under cultivation conditions.

At the same time, it is argued that only Carbonactinospora thermoautotrophica are vital for the survival of all consortium.

A: Thank you for your constructive assessment and positive feedback. You are right and we have restructured and rephrased several parts of the text to better reflect our results and refine the breadwinner idea. In this sense, we now include some caveats and highlight specific interactions and dependencies, such as:

Lines 582-586: “The loss of *C. thermoautotrophica* StC would be fatal to the other members of this consortium. The other members would be the so-called “beneficiaries,” where *Chelatococcus* spp. and *Sphaerobacter* spp. would be the “collaborators” (members that are non-essential but provide some benefits to the group), and *Geobacillus* spp. would be an “opportunist” or “cheater”, at least considering the available data and energetic requirements.

Lines 602-605: “...However, it is reliable to state that all beneficiaries are completely dependent on the energy provided by the breadwinner *C. thermoautotrophica* StC, whereas the input from other members could contribute to the efficiency of the consortium’s survival in such a nutrient-depleted medium.”

Lines 537- 540: “From the consortium, only *Geobacillus* spp. (LEMMY01) could be isolated using a low nutrient Reasoner’s 2A medium (R2A), which is one of the most used media for the isolation and growth of oligotrophic, environmental bacteria. However, the same strain was incapable of growing axenically in an N-FIX medium with a gaseous atmosphere.”

Also, the work does not demonstrate that the exclusion of one of the secondary members of the community leads to the impossibility of its growth under cultivation conditions.

A: Indeed, this assumption is based on the MAGs and *Geobacillus* spp. genome obtained, known metabolic traits and their complementarities. It is a hypothesis based on the available data and known metabolic traits. We have included some data that can back up this idea and better organized this line of thinking to also include caveats and limitations, as shared in our previous response and throughout the discussion, as for example:

Lines 464-485: “Following phylogenetic analysis, the metabolic potential of the consortium members was investigated. All four organisms harbored genes encoding proteins that facilitated aerobic respiration; this finding was consistent with the results obtained in our experimentally created aerated growth conditions. Meanwhile, genes required for the oxidation of organic carbon (heterotrophic metabolism) were only detected in *Sphaerobacter* spp., *Chelatococcus* spp., and *Geobacillus* spp. The high-affinity lineages identified in *C. thermoautotrophica* StC support its ability to scavenge H₂ and CO present in trace concentrations in the atmosphere (Mulkan et al., 2017) In addition, *C. thermoautotrophica* StC MAG harbored genetic information required for carboxydutrophy (*coxS*, *coxM*, and *coxL*) and autotrophy through type IE RuBisCO, which is a member of the CBB cycle. Although no genes related to CO₂ fixation were detected in *Chelatococcus* spp. and *Sphaerobacter* spp., they possessed complete pathways for oxidizing CO and could obtain energy from this trace gas. Notably, the expression of group 1 hydrogenase detected in *C. thermoautotrophica* StC and *Sphaerobacter* spp. and group 2a detected in *C. thermoautotrophica* StC could facilitate the use of H₂, another trace gas, as an energy source by these microbes. Cumulatively, our data indicated that *C. thermoautotrophica* StC primarily supplies carbon to the microbial consortium through carbon fixation. The energy for the system is provided by *C.*

thermoautotrophica StC, *Chelatococcus* spp., and *Sphaerobacter* spp. as they appear to be able to use CO as an energy source. The metabolic compartmentalization of these species, along with the inability to isolate individual components among them, indicates some level of interdependence (at least from some members) and collaboration among the taxa, whereas *Geobacillus* spp. probably benefits from organic compounds produced by the other members in such oligotrophic systems.”

Lines 587-605: “...Further studies using gene expression assays are required to compare gene expression and metabolomics in both the absence and presence of nitrogen to help elucidate the mechanisms underlying the survival of this consortium. Despite the evidence regarding incomplete nitrogen assimilation pathways, we cannot rule out the possibility of the existence of other “breadwinners” in the consortium. The nitrogen assimilation mechanism is still unclear and could be a key component associated with specific members of the consortium. Therefore, the isolation of other members of the consortium, in addition to *Geobacillus* sp., may be useful to clarify key mechanism(s) via in vitro assays; compared with prediction using metagenomic data, such assays are more stable and facilitate the study of biochemical functions with greater accuracy. For example, several apoenzymes involved in carbon and nitrogen metabolism exhibit post-translational modifications and cofactor maturation (such as molybdopterin and Fe-heme) (Blake et al. 2022). Isolation of the main component—*C. thermoautotrophica* StC—would also allow us to examine, model, manipulate, and “design” metabolic pathways in natural and constructed microbial systems using systemic approaches and synthetic biology. However, it is reliable to state that all beneficiaries are completely dependent on the energy provided by the breadwinner *C. thermoautotrophica* StC, whereas the input from other members could contribute to the efficiency of the consortium’s survival in such a nutrient-depleted medium.”

The Introduction is well-written, with clear objectives for the study, but I suppose one or two short examples of specific "novel biotechnological applications" or "novel bioproducts" might be a good addition.

A: Thank you. We have included some specific examples, as follows:

Lines 56-64: “A better understanding of this interdependence and interaction modes driving microbial community interactions (Tan et al., 2015) may lead to the discovery of novel pathways or genes (Narayanasamy et al., 2015) and eventually to novel biotechnological applications (Lindemann et al., 2016; Jawed, Yazdani, and Koffas 2019) through the optimization and engineering of microbial consortia for the production of biomolecules, biofuels and carbon sequestration (Jones et al. 2017; Liu et al. 2018; Correa et al., 2022), and by the use of the principles of microbial ecology (Koch, Korth, and Harnisch 2018). These advances may contribute for the mitigation of crucial issues, such as climate change and population growth (Lindemann et al., 2016; Santoro et al., 2021; Peixoto et al., 2022).

The Methods and Results are generally very detailed and descriptive (see exceptions below).

Line 115, 142 Need to clarify - were the samples taken aseptically and the gases filtered?

A: Thank you for pointing this out, this is indeed crucial information that was missing. We have included the following sentences in the method section:

Line 90: "Soil was collected using spatulas disinfected with 70% ethanol."

And

Lines 120-122: "...The gas solution applied was previously collected from cannisters using a sterile syringe and filtered in Millipore syringe filters (0.2 μm). Sterile needles were also used throughout the entire process."

Line 125 The cited reference (Embrapa, 1997) is not listed in References

A: This part of the text was restructured and more accurate references were included, as Embrapa, 1997 is in Portuguese.

Line 133 Here, the abbreviation of the medium is given as NFIX, while the spelling N-FIX is also implied later in the text.

A: Good catch. We have standardized it to N-FIX throughout the text.

Line 134-138 Please check if the rules of the journal allow the indication of concentrations in the form g/L, mg/L (I suppose unit dimensions should be expressed using negative integers). $\text{CaCl}_2 \cdot \text{H}_2\text{O}$ is it $\text{CaCl}_2 \cdot 2\text{H}_2\text{O}$? What was the pH of the medium?

A: We have checked the entire text (which was double checked by the proofreading service team) and standardized these units according to the guidelines provided at <https://www.nature.com/documents/commsj-life-style-formatting-guide-accept.pdf>

We have also fixed the Calcium chloride formula. The medium pH was adjusted to 7.2

Line 180 The final concentration of SDS in the lysis buffer should be indicated.

A: Good point. This has been included, as follows:

Lines 175-177: "To improve cell recovery, 1 mL of 10% (volume/mass of approximately 0.03 M) sodium dodecyl sulfate solution was added to each flask

containing culture media, with a final concentration of approximately 0.25% (0.0008 M).”

Line 190 Here it is necessary to give a decryption of nifH

A: This has been added, as follows:

Lines 189- 191: “...determined using PCR by targeting the *nifH*—a gene that encodes the dinitrogenase reductase subunit of the nitrogenase enzyme, which is a biological marker for nitrogen fixation (Gaby and Buckley 2014).”

Line 198-212 Were the sequences resulting from the DGGE deposited in the NSBI database or others?

A: Our goal with the DGGE was to observe the consortium stability over time, based on the presence and absence of bands, but these bands were not sequenced.

Line 202 Here it is necessary to give a decryption of rrs.

A: Included, as follows:

Lines 205- 207: “...a hypervariable region of the gene *rrs* was amplified, and gene codification of the ribosomal RNA composing the small subunit of the bacterial ribosome (16S) was performed (Heuer et al., 1997).”

Line 316-321 Why was the tactic of culturing the community on different media but not antibiotic treatment chosen, taking into account the presence of both Gram-positive and Gram-negative bacteria in the consortium?

A: This is a very good suggestion and, although it was not the goal of this first study, which is already very comprehensive, is now subject of our further studies. We now discuss this at the end of our manuscript, as follows:

Lines 592- 605: “Therefore, the isolation of other members of the consortium, in addition to *Geobacillus* sp., (as, for example, using specific antibiotics and culture conditions, based on the data obtained from the MAGs) may be useful to clarify key mechanism(s) via in vitro assays;...”

Line 339-341 Here the authors indicate that "(8% of reads) were related to Chelativorans" however, the rest of the text refers to a member of the genus Chelatococcus. It here and in Fig2 is also necessary to give the correct names of the phyla.

A: These edits have been included. This is now Fig. 3.

Figure 3

Extended Data Figure 1 The table needs careful editing

A: Thank you. This table has been fully edited and is now Supplementary Figure 1.

Parameter	Methods	Unit	Control Soil	Burned soil
pH	(H ₂ O)	NA	7.4	8.1
Total N	Dumas	%	0.075	0.059
Organic C	(Oxi-Red)	%	1.5	1.8
M.O	(Oxi-Red)	dag/kg ²	2.5	3.1
C/N Ratio	NA	NA	19.5	30.7
P	(Mehlich-1)	mg/dm ³	346	977
K	(Mehlich-1)	mg/dm ³	44	778
Na	(Mehlich-1)	cmolc/dm ³	0.21	0.29
Ca	(KCl 1 mol/L)	cmolc/dm ³	6.9	5.7
Mg	(KCl 1 mol/L)	cmolc/dm ³	1.8	2.7
K	(KCl 1 mol/L)	cmolc/dm ³	0.11	1.99
Al	(KCl 1 mol/L)	cmolc/dm ³	0	0
H+Al	Ca Acetate	cmolc/dm ³	0.4	0.1
S	Sum of bases	cmolc/dm ³	9	10.7
T	(C.T.C.)	cmolc/dm ³	9	10.8
V%	Saturation of bases	cmolc/dm ³	95.6	99
m	Al saturation	%	0	0
n	Na saturation	%	2.2	2.7
t	effectiv C.T.C.	%	9	10.7

Supplementary Figure 1. Field site where the samples were collected. Soil samples were collected from under the pile of burned vegetal material. Images of the typical burning process in the sampling site (A and B) and a table showing the soil characterization under the vegetal ashes and in a control soil with no history of burning collected nearby (C).

Discussion

In general, the authors presented a good discussion of the results obtained using modern methods. It is obvious that additional experimental studies are required to elucidate all aspects of the functioning of the described community. In particular,

both the omics approach and chemical analysis of the environment in order to identify metabolites exchanged among community members, as well as possible sources of nitrogen. It would also be interesting to know how significant the role of such a consortium is in the natural community of the studied soil. I hope this will become a task for future work.

A: Thank you again for the very constructive feedback and ideas. We have restructured this section and included a more comprehensive discussion about our data and future work and challenges. We have already started exploring these aspects in ongoing long-term projects to test these concepts.

REVIEWERS' COMMENTS:

Reviewer #2 (Remarks to the Author):

The authors have done an excellent job in improving their previous manuscript on the idea of the breadwinner hypothesis. However, since the nitrogen assimilation mechanism in the community is still unclear, nitrogen could also be playing the role of "the bread" thus the possibility of multiple breadwinners.

The authors have acknowledged this limitation in their manuscript and have suggested further research to clarify the nitrogen assimilation mechanism and the possibility of multiple breadwinners. They have also highlighted the need for gene expression assays to compare gene expression and metabolomics in both the absence and presence of nitrogen to help elucidate the mechanisms underlying the survival of this consortium. Furthermore, they have also suggested the isolation of other members of the consortium to clarify key mechanisms via in vitro assays. This will provide more stable and accurate results compared to prediction using metagenomic data.

In conclusion, the authors have successfully improved the readability and comprehensibility of the manuscript and have successfully addressed the revisions proposed in the previous version. They have acknowledged the limitations of their research and have provided suggestions for further research to clarify key mechanisms and the possibility of multiple breadwinners. Their proposed idea of the breadwinner hypothesis is an important contribution to the understanding of biological systems.

Reviewer #3 (Remarks to the Author):

The submitted manuscript is the revised version of the primary draft.

The authors made a great effort to improve the manuscript and considered the comments and feedback from the review. This version of the manuscript has become more apparent and answered most of the comments from the first round.

I have a one minor remark that do not affect the overall positive impression of this article:

The contamination of one of the MAGs is 12%. In my opinion, it is better to perform its manual refinement. However, I think this will not affect the results of the work.

Here is a point-by-point response to the reviewers' comments and concerns.

Author responses are in **blue**.

Comments from Reviewer #2 (Remarks to the Author):

The authors have done an excellent job in improving their previous manuscript on the idea of the breadwinner hypothesis. However, since the nitrogen assimilation mechanism in the community is still unclear, nitrogen could also be playing the role of "the bread" thus the possibility of multiple breadwinners.

The authors have acknowledged this limitation in their manuscript and have suggested further research to clarify the nitrogen assimilation mechanism and the possibility of multiple breadwinners. They have also highlighted the need for gene expression assays to compare gene expression and metabolomics in both the absence and presence of nitrogen to help elucidate the mechanisms underlying the survival of this consortium. Furthermore, they have also suggested the isolation of other members of the consortium to clarify key mechanisms via in vitro assays. This will provide more stable and accurate results compared to prediction using metagenomic data.

In conclusion, the authors have successfully improved the readability and comprehensibility of the manuscript and have successfully addressed the revisions proposed in the previous version. They have acknowledged the limitations of their research and have provided suggestions for further research to clarify key mechanisms and the possibility of multiple breadwinners. Their proposed idea of the breadwinner hypothesis is an important contribution to the understanding of biological systems.

We appreciate the Reviewer's input to review the revised version and give this positive comment.

Reviewer #3 (Remarks to the Author):

The submitted manuscript is the revised version of the primary draft. The authors made a great effort to improve the manuscript and considered the comments and feedback from the review. This version of the manuscript has become more apparent and answered most of the comments from the first round. I have a one minor remark that do not affect the overall positive impression of this article:

The contamination of one of the MAGs is 12%. In my opinion, it is better to perform its

manual refinement. However, I think this will not affect the results of the work.

Response: We apologize for the wrong data/information. Unfortunately, there was a confusion with the correct accession number in the manuscript.

- Original: "Raw reads were deposited in the NCBI Sequence Read Archive under the BioSample accession number SAMN15590030."

This accession number refers to all members of the consortium. What the reviewer found as contamination is actually the other minority members of the consortium. A new sentence was added to the manuscript with the correct accession number, containing only the MAG without contamination. Nucleotide sequences from biosample SAMN07787832 were used in all analysis and figures of *C. thermoautotrophica* MAGs, without contaminations.

Accordingly, we have now revised the Discussion section as the following:

- Original: "Raw reads were deposited in the NCBI Sequence Read Archive under the BioSample accession number SAMN15590030."

- Revised: Raw reads of the consortium sample were deposited in the NCBI Sequence Read Archive under accession number SRR12323727. The *Carbonactinospora thermoautotrophica* MAG contigs were deposited in the NCBI under the WGS master record PQID00000000.1 and Biosample SAMN07787832.

We thank for this opportunity to clarify this point.